# Spectral replacement using machine learning methods for continuous mapping of Geostationary Environment Monitoring Spectrometer (GEMS)

Yeeun Lee[1], Myoung-Hwan Ahn[1], Mina Kang[1], Mijin Eo[1]

[1]Department of Climate and Energy Systems Engineering, Ewha Womans University, Seoul, 03760, Republic of Korea

*Correspondence to*: Myoung-Hwan Ahn (terryahn65@ewha.ac.kr)

**Abstract.** Earth radiances in the form of hyperspectral measurements contain useful information on atmospheric constituents and aerosol properties. The Geostationary Environment Monitoring Spectrometer (GEMS) is an environmental sensor measuring such hyperspectral data in the ultraviolet and visible (UV/VIS) spectral range over the Asia-Pacific region. After

completion of the in orbit test (IOT) of GEMS in October 2020, bad pixels are found as one of remaining calibration issues because during the IOT one-dimensional interpolation performed to replace the erroneous pixels caused high interpolation error for a wide defective area on the detector array. The error results in obvious spatial gaps in the measured radiances and also retrieved properties. To resolve the fundamental cause of the issue, this study takes an approach reproducing erroneous spectral features of the defective spectra with machine learning methods with artificial neural network (ANN) and multivariate

linear regression (Linear). The basic assumption of the methods is that radiances in a spectrum have linear and non-linear relations and a finite range of radiances can be reproduced with the relations. The machine learning models are trained with defect-free measurements of GEMS after dimensionality reduction with principal component analysis (PCA) for efficient model training. Results show that PCA-Linear has small prediction errors especially for a narrower spectral gap and less vulnerable to outliers with an error of 0.5-5%. PCA-ANN shows better results emulating strong non-linear relations with an

error of about 5% except for the shorter wavelengths around 300 nm. It is verified that dominant spectral patterns can be successfully reproduced with the models within the level of radiometric calibration accuracy of GEMS, but a limitation still remains when it comes to much finer spectral features. When applying the reproduced spectra to retrieval processes of cloud and ozone, the cloud centroid pressure shows an error of around 1% while total ozone column density shows relatively higher variance. It seems to be clear that the effectiveness of the method can be improved further by optimally setting the input and

output spectral bands for the spectral replacement. As an initial step in reproducing spectral patterns for erroneous spectra, this study verifies machine learning methods have high potential to be updated further for enhancing measurement quality of environmental satellite measurements.

# 1 Introduction

Earth radiances can provide useful information on the atmospheric chemical composition, especially when it is measured in
the form of many contiguous spectral bands. This type of measurements is referred to as 'hyperspectral' (Bovensmann et al.,
1999; Goetz et al., 1985) which is frequently sampled with high spectral resolution to accurately describe absorption lines of
targeted gaseous or particulate components (Boersma et al., 2004; Kang et al., 2020; Manolakis et al., 2019; Pan et al., 2017).
The Geostationary Environment Monitoring Spectrometer (GEMS) on-board the Geostationary Korea Multi-Purpose Satellite-
2B (GEO-KOMPSAT-2B) is an environmental sensor providing such a hyperspectral measurement in the ultraviolet and
visible (UV/VIS) spectral region from 300 to 500 nm with a spectral resolution of finer than 0.6 nm (Kim et al., 2020).
Following the launch of the satellite in February 2020, the in orbit test (IOT) of GEMS was successfully completed in October
2020 with some issues to be continuously monitored. The root cause of each issue is to be examined with collected long-term
measurements on the radiance level (Level 1B), as it has been dealt with for other polar orbit satellite sensors having similar
sensor characteristics (Ludewig et al., 2020; Pan et al., 2019, 2020; Schenkeveld et al., 2017).

40          One of the issues to be periodically monitored is bad pixels, which refer to anomalous pixels having hot, cold, noisy
or drifted readout values in raw data (Ló´pez-Alonso and Alda, 2002). The definition of bad pixels is not universal, and in this
paper, it refers to all kinds of pixels having abnormal observation features. Bad pixel detection is based on the sensor
characterization sorting out erroneous signals from a normal trend and a few hot pixels were flagged as bad pixels during on-
ground tests for GEMS. Additionally, more pixels have been sorted out during the IOT because of the impacts from the launch
of the satellite and different environment conditions in space. The number of bad pixels may increase as time goes by (Kieffer,
1996), which indicates a significant number of bad pixels could affect measurement quality during the operation period.

          Subsequently, the detected pixels should be replaced (Boldrini et al., 2012; Burger, 2009; Rankin et al., 2018) and in
the GEMS calibration system, it adopts spatial interpolation on the detector array along the spatial direction (Fischer et al.,
2007; Schläpfer et al., 2007). However, the approach showed its limitation during the IOT, when an area consisting of bad
pixels is quite large and the adjacent pixels valid for spatial interpolation are too far from the erroneous area. Especially, when
a scene on the Earth dramatically changes, discontinuity caused by the interpolation becomes more apparent. This effect causes
spatial discontinuity in Level 1B data and retrieved properties (Level 2) by affecting retrieval processes with contaminated
spectral features.

          As a way of filling in the spatial gaps, this study approaches the underlying problem by focusing on radiances with
spectral replacement using machine learning methods. The spatial gaps found in Level 2 data can be filled in with various
methods (e.g. variogram, empirical orthogonal functions or mathematical filters) and for each Level 2 product, there will be a
more suitable method using multiple sources of information and distribution characteristics (Fang et al., 2008; Guo et al., 2015;
Katzfuss and Cressie, 2011; Llamas et al., 2020; Yang et al., 2021). In this regard, this research places more emphasis on
efficiency and further application of the approach because improving erroneous spectral features can be an efficient way to
solve the issue for all products and also has the potential to be applied to various measurement issues of hyperspectral data.

For that, further questions to be investigated here are whether non-linear relations could be accurately emulated with machine learning methods and input radiances have valid information for retrieval processes. For the investigation, cloud and ozone retrievals are performed with the reproduced spectra of GEMS to evaluate the effectiveness of the suggested approach and its limitations.

In this respect, we suggest machine learning models with artificial neural network (ANN) and multivariate linear regression which is trained to emulate spectral relations. Theoretically, it has been verified that ANN can accurately emulate non-linear relations with a simple model structure when there are a large number of training data (Cybenko, 1989; Hornik et al., 1989). Machine learning methods also have a high chance to successfully process hyperspectral data because the abundant datasets make the training process more efficient after breaking the curse of dimensionality with a proper pre-processing step

(Gewali et al., 2018). For that, principle component analysis (PCA) is applied in this study, as it has been known to be very useful to extract important information from hyperspectral measurements (Bajorski, 2011) and widely used to retrieve environmental and surface properties (Horler and Ahern, 1986; Joiner et al., 2016; Li et al., 2013, 2015).

For atmospheric remote sensing, the majority of researches has employed machine learning as a proxy of the radiative transfer model to retrieve geophysical states from measured spectral radiances (Hedelt et al., 2019; Loyola et al., 2018; Zhu et

al., 2018). There are fewer approaches applied to obtain radiation flux (Dorvlo et al., 2002; Zarzalejo et al., 2005) and even much fewer to obtain hyperspectral radiances for different purposes such as to accurately quantify radiative forcing in climate system (Taylor et al., 2016), increase spectral resolution (Le et al., 2020) and fill in a spectral gap for inter-calibration (Wu et al., 2018). A monochromatic radiance itself rarely contains any important meaning and thus seldom has it been a final target for machine learning. In this study, however, radiance at each wavelength of a targeted spectral region are an important output

to be reproduced.

The following sections are organized as follows. Section 2 introduces sensor specification of GEMS and a general description of machine learning models with model structure and hyperparameter setting. Section 3 contains model optimization results and error analysis for wide defect regions. With the optimized model, the spatial and spectral inspection is performed for reproduced radiances and retrieved properties. In Sect. 4, conclusions are presented with limitations as well

as further application of the method in future study.

## 2 Data and methods

### 2.1 Data description

#### 2.1.1 GEMS

GEMS is a UV/VIS imaging spectrometer in the geostationary orbit observing the Asia-Pacific region (5° S-45° N, 75° E-145°

E) with high spatial and spectral resolution to retrieve key atmospheric constituents such as ozone ($O_3$), sulfur dioxide ($SO_2$), nitrogen dioxide ($NO_2$), formaldehyde (HCHO), glyoxal (CHOCHO) and aerosol properties  (Kim et al., 2020). The

observation targets of GEMS are the Sun (irradiance mode) and the Earth (radiance mode) and the description for each measurement mode is summarized in Table 1. In both measurement modes, incident light from a scene passing through a fore-optics and a spectrometer reaches to a two-dimensional detector array, the charge-coupled device (CCD) detector. The CCD of GEMS comprises 2,048 rows and 1,033 columns of photoactive pixels along the spatial direction from north to south and the spectral direction with a sampling interval of 0.2 nm, respectively. GEMS observes the Sun on the purpose of calibration once a day with a premise for the measured solar irradiance being stable and nearly time independent. For Earth measurements, GEMS measures the backscattered radiation from east to west about 700 times by moving a scan mirror and for each scan, totally 2048 pixels are obtained along the north-south direction. All measurements at each scan position are combined together to cover the full field of regard (FOR) of GEMS. The data used in this study are the operational data (Level 1C) which are used for the retrieval processes of Level 2 products.

**Table 1** Top level measurement specifications of GEMS

| Measurement mode | Solar irradiance | Earth radiance |
|---|---|---|
| Data dimension [spectral, spatial, scan] | [1033, 2048] | [1033, 2048, 695] (nominal scene) |
| Spectral range [nm] | 300-500 | |
| Spectral sampling [nm/pixel] | 0.20 | |
| Spectral resolution [nm] | < 0.60 | |
| Spatial resolution [km$^2$] | - | $3.5 \times 8$ (spatial $\times$ scan) |
| Measurement frequency | Once a day (13:00 UTC) | Every hour (00:45-07:45 UTC) |

### 2.1.2 Bad pixel

Bad pixel detection is generally performed with dark-current measurements which are taken without exposure to light for a certain integration time (Howell, 2006), and for GEMS, the integration time corresponds to about 70 milliseconds. Figure 1 illustrates bad pixel positions (in white) on the GEMS CCD detector array identified during the IOT. A cluster and distinct line shapes of bad pixels shown in Fig. 1a are initially identified during on-ground calibration before the launch of the satellite. Following the suggestions made by the instrument developers, linear interpolation along the spatial direction (north-south) is applied to replace the unusable measurements on bad pixel positions and a single bad pixel could be properly substituted with such a simple procedure. However, it was found during the IOT that significant interpolation error could occur on the bad pixel positions denoted as Defects 1-3 (see Fig. 1b), especially when the invalid spatial width is too wide such as Defects 2-3.

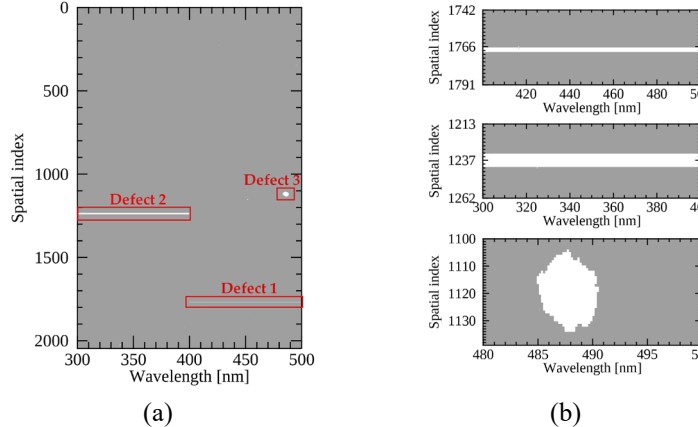

(a)                  (b)

**Figure 1** The two-dimensional bad pixel map (a) on the GEMS CCD detector along the spectral (x-axis) and spatial direction (y-axis) and (b) zooming in the bad pixel positions from top to bottom rows for Defects 1-3. Bad pixels are marked in white.

The interpolation error seriously affects Level 2 products of which the spectral fitting windows are overlapped with bad pixel areas. For instance, cloud properties and aerosol effective height (AEH) of GEMS are retrieved from $O_2$-$O_2$ absorption bands around 477 nm (Choi et al., 2021; Kim et al., 2021) where the cluster of bad pixels is located (Defect 3). During the IOT, Defect 3 caused spatial discontinuity to the retrieved cloud and AEH distribution, which made the fitting window of the products modified to avoid bad pixel effects. Ozone retrieval is also affected by Defect 2 (300-400 nm) as the spectral radiances within 300-380 nm are major ozone absorption lines in the UV/VIS spectral range (Bak et al., 2019). Even though spatially interpolated radiances are homogeneous with its surroundings (see Fig. 2), the spectral patterns are not properly reproduced with the existed method (spatial interpolation) which causes distinct horizontal lines to the retrieved products, to be discussed in Sect. 3.2 2.

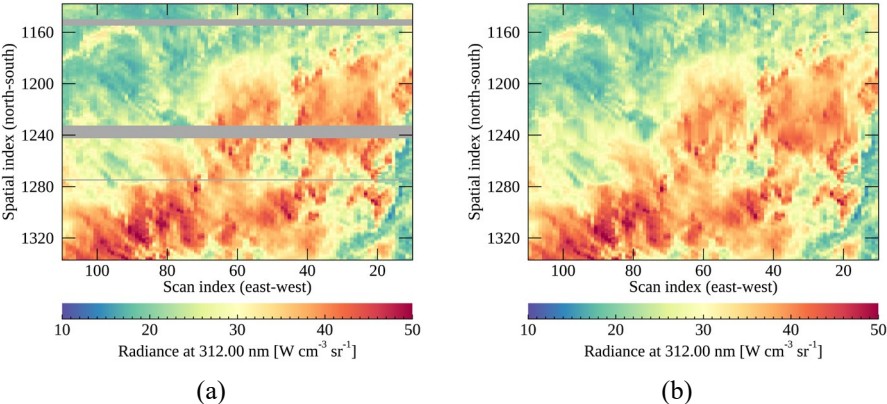

(a)                  (b)

**Figure 2** Spatial distribution of GEMS radiances at 312 nm with bad pixels (a) marked in dark gray and (b) reproduced with spatial interpolation. The GEMS spectra are measured on 10 March 2021 (06 UTC).

## 2.2 Replacement approach

### 2.2.1 General description

Upwelling radiances are determined by the interactions of light with trace gases, aerosols and clouds in the atmosphere and surface reflectivity. Especially, scene properties are the dominant factor resulting in strong linear relations among radiances in a spectrum. In other words, when a scene is dark (bright), the upwelling radiances of the scene over the whole spectral region tend to become generally low (high). Spectral replacement is based on a fact that radiances at different wavelengths for a scene are highly correlated with each other and have certain spectral relations (Liu et al., 2006; Wu et al., 2018). If the relations are accurately emulated, some missing values in a spectrum can be reproduced with radiances at the other wavelengths. The important question here is whether non-linear relations can be accurately reproduced with only the spectral information. To investigate this, randomly collected GEMS spectra measured on defect-free pixels are used to establish the relations with the basic premise that neighbor pixels on the detector array (set to within 100 spatial indices) would have similar measurement characteristics.

Because it is highly possible that input radiances have redundant information, PCA is applied for dimensionality reduction to compress the input radiances to low-dimensional principle components (PCs). The strong linear relations among radiances in a spectrum are compressed to the first PC, which has the largest variance. The non-linear properties caused by atmospheric scattering, absorption, different optical paths and sensor noise are projected onto the subsequent PC subspaces. The PCA process is given by the following Eq. (1):

$$\mathbf{Z}_{n \times p} = \mathbf{X}_{n \times \lambda} \mathbf{W}_{\lambda \times p} \tag{1}$$

where $\mathbf{Z}$, $\mathbf{X}$ and $\mathbf{W}$ represent the PC scores, input and PC matrix, respectively. The PC scores matrix ($\mathbf{Z}$) is obtained by projecting the input to the PC subspaces with $\mathbf{W}$, which is obtained by applying eigenvalue decomposition to the $\mathbf{X}$. The subscript $n$, $\lambda$ and $p$ indicates the dimension of matrix corresponding to the number of datasets, input wavelengths and the number of PCs, respectively.

With the compressed data, multivariate linear regression (PCA-Linear) and ANN (PCA-ANN) models are trained to define the relations between input ($\mathbf{X_m}$) and output ($\mathbf{Y_n}$) radiances in a spectrum. The PCA-ANN model is constructed with a simple feed-forward model with a hidden layer as described in Fig. 3. In the model optimization process, the PCA-ANN model with a hidden layer showed faster and more effective convergence of loss function than the models having multi-hidden layers in this study. For PCA-Linear, it adopts a simple linear model structure consisting of parameters such as weight and bias having the minimum mean squared error (MSE) between the regressed and measured radiances. After model optimization, it can be used to replace bad pixels ($\mathbf{X'_m}$, $\mathbf{Y'_n}$) with reproduced radiances likely measured by the sensor.

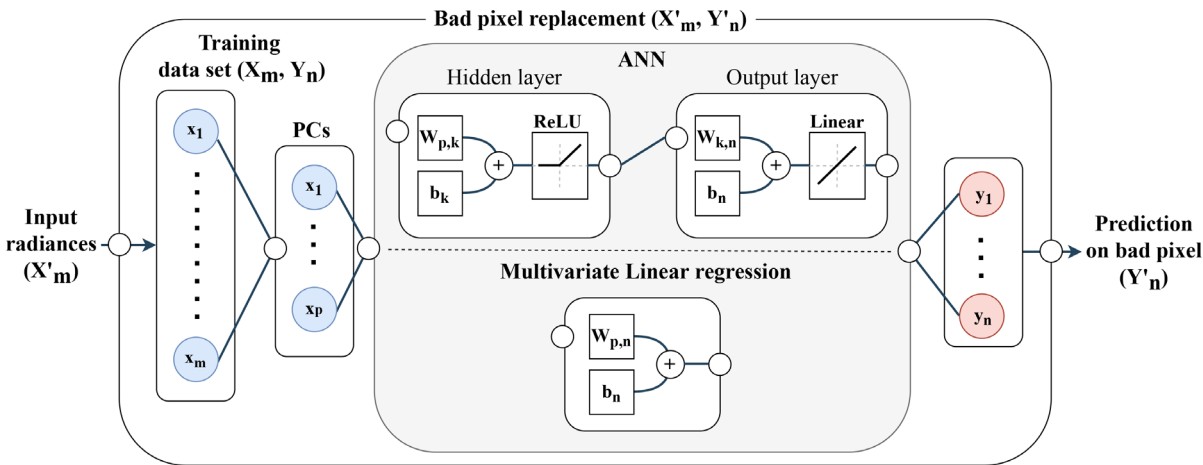

**Figure 3** Schematic chart of the training and bad pixel replacement process. **W** and **b** represent weight and bias parameters in each layer. The subscript *m, n, p* and *k* is equal to the spectral dimension of input and output parameters, the number of PCs and hidden nodes of the ANN model, respectively.

### 2.2.2 Input/output and model optimization

For the model training, radiances of each spectrum are divided into input and output radiances based on the specified spectral

ranges in Table 2. The spectral ranges of output radiances for Defects 1-3 are identical to each defective region and the rest part of a spectrum becomes input radiances. The constructed input and output datasets are split into training and test data to update model parameters and check for overfitting, respectively which are randomly sampled out in March 2021. When training a machine learning model, it is important to sample the similar numbers of bright or dark scenes because the majority corresponds to dark scenes with random selection. Considering that the training process becomes unstable when the collected

scenes are skewed to low radiances, oversampling is a necessary step before the scene selection by binning the datasets depending on the magnitude of spectra.

The datasets for the models should be sampled at identical spectral grids and for that, each spectrum is interpolated in a pre-processing step and after the reproduction, the spectra are reversely interpolated onto its original spectral grids. Considering that the intrinsic information a spectrum has could be lost during the interpolation processes, finer spectral grids

(0.1 nm) are adopted for the model to minimize interpolation errors by preserving radiances at more frequent intervals than the original grids. The solar zenith angle (SZA) and viewing zenith angle (VZA) are key variables determining optical paths of upwelling and downwelling radiances and thus are used as input variables together with radiances.

The neural network constructed with the hyperparameter setting presented in Table 2 is implemented with TensorFlow, a high-level Application Programming Interface (API) written in Python. As described in Fig. 4, the activation

function is the Rectified Linear Unit (ReLU) in the hidden layer of the ANN model. The structure itself is not complicated but it has multiple nodes in the input and output layers, which makes ReLU more competitive (Nwankpa et al., 2018). The

hyperbolic tangent (tanh) and sigmoid function show poor results especially when the output parameters have lower variance making the optimization stuck into the averaged value and preventing the model from being updated.

Table 2 Input and output (I/O) parameters for ANN training and hyperparameter for optimization of neural network.

| Model | Parameter | Defect 1 | Defect 2 | Defect 3 | Remark |
|---|---|---|---|---|---|
| I/O | Input ($X_m$) | SZA / VZA | | 460-483.9 / 491.1-500 nm | Random selection (100,000 for training and test data) |
| | | 300-400 nm | 400-500 nm | | |
| | Output ($Y_n$) | 400.1-500 nm | 300-399.9 nm | 484-491 nm | |
| Hyper-parameter | Activation function | ReLU | | | |
| | Optimizer | Adam optimizer | | | |
| | Loss function | Mean squared error | | | |
| | Scaling | Standardization | | | |


For the optimizer, Adaptive Moment Estimation (Adam) is used which shows stable results compared to Stochastic Gradient Descent (SGD) and Root Mean Square Propagation (RMSProp) (Kingma and Ba, 2015). It is empirically found that SGD without gradient clipping tends to cause exploding gradient and RMSProp has difficulty reaching the global minima compared to Adam. Figure 5 presents the converging process of the PCA-ANN model for Defect 2 applying different

optimizers with and without SZA and VZA conditions. The addition of the angle conditions as input parameters speeds up the model convergence with smaller MSE because without the angle parameters, the information would be implicitly elicited during the optimization process. The model converges with angle conditions at 44, 98 and 33 epochs for Adam, SGD and RMSprop, respectively. Adam converges at the smallest MSE while SGD converges with the highest MSE. RMSprop presents unstable loss for validation data and converges with higher MSE compared to Adam.

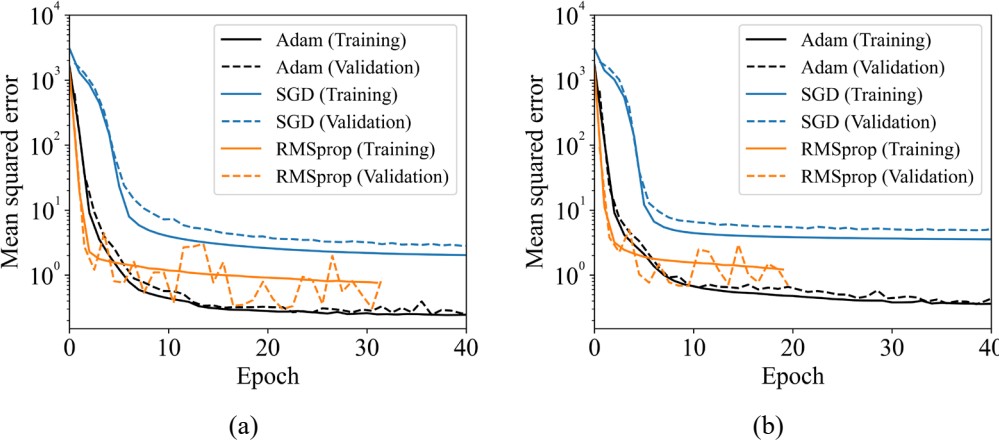

(a)                                                          (b)

**Figure 4** Training and validation losses for Defect 2 (a) with and (b) without the angle conditions as input parameters. The results are obtained with different optimizers such as Adam (black), SGD with the gradient clipping value of 0.5 (blue) and RMSprop (orange).

# 3 Results and discussion

## 3.1 Model selection

### 3.1.1 Optimization results

Figure 5 shows model optimization results depending on each model and the number of PCs as the input nodes. Because the spectral range of output radiances differs for each defect region (Defects 1-3), model optimization is also performed, respectively. The spectral ranges of output radiances for Defects 1 and 2 are wider than that of Defect 3 which results in higher MSE. PCA-ANN seems to be unstable for Defect 1 showing overfitting which might be caused by unfiltered outliers in output radiances of GEMS at the wavelengths longer than 480 nm. Defect 2 is at the wavelengths where the upwelling radiances are

largely affected by ozone, which increases non-linearity between input and output radiances. Because of the strong non-linearity, PCA-ANN shows better performance than PCA-Linear for Defect 2. Defect 3 has the smallest number of output parameters in a narrow spectral gap which causes strong correlation between input and output radiances. The loss functions (MSE) in Fig. 5c are small and converge quickly for both PCA-ANN and PCA-Linear models. With the results, the optimized number of PCs is set to 90 for all defect regions when loss functions for both training and test data efficiently converge, with

PCA-Linear for Defects 1 and 3 and the PCA-ANN model for Defect 2.

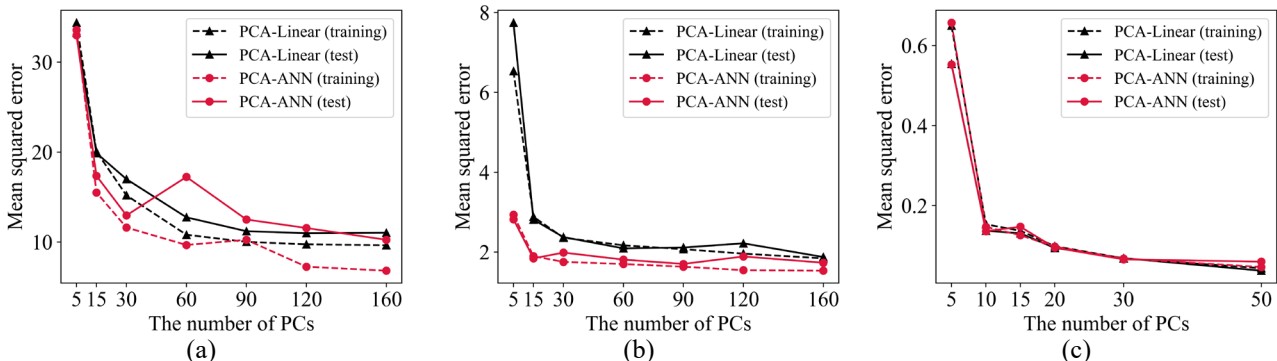

**Figure 5** Loss function depending on the number of PCs with PCA-ANN (red) and PCA-Linear (black) models for spectral replacement with training and test datasets for Defects 1-3 ((a): Defect 1, (b): Defect 2 and (c): Defect 3). The number of hidden nodes for ANN is double the number of PCs.

### 3.1.2 Statistical evaluation

The optimized model structures for Defects 1-3 are set as described in the previous section. Following that, in this section, model performance is statistically evaluated with training and test datasets specified in Table 2. Figure 6 presents mean and normalized root mean squared error (NRMSE) of the output radiances of training and test data. The NRMSE is a statistical indicator normalized by the mean radiance at each wavelength and it can be found that radiances at strong absorption lines have relatively high uncertainty. Especially, information from the radiances in 400-500 nm is insufficient to properly represent

ozone absorption features at shorter wavelengths and it causes high uncertainty at the wavelengths shorter than 325 nm in Defect 2. Defect 1 also has higher errors around the edges of output spectral ranges where pixel saturation could be found. It

is clear that prediction errors of Defects 1-2 increase at the output wavelengths far from the input spectral bands. Defect 3 shows the smallest NRMSE of around 0.2% because of strong linear relations between input and output radiances as previously mentioned in Sect. 3.1. The results show that it is possible to successfully reproduce spectral features at a narrower spectral range with simple linear regression.

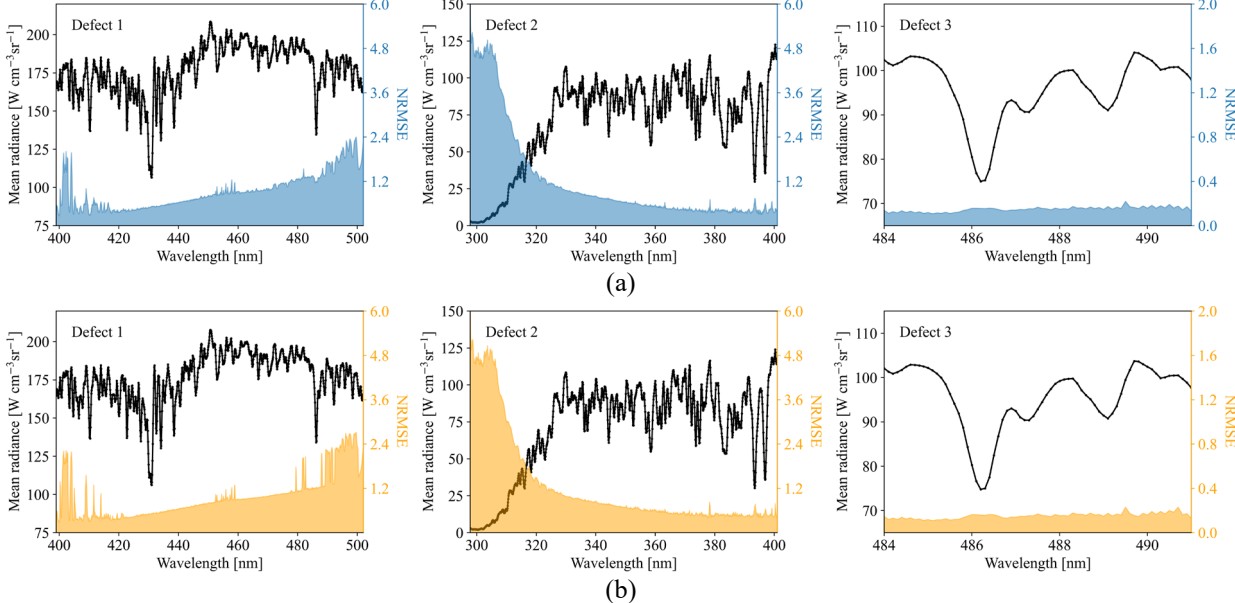

**Figure 6** Output radiances for Defect 1-3 with averaged spectra and NRMSE for (a) training and (b) test datasets measured in March 2021. The unit of NRMSE is in percent.

Figure 7 presents error histograms of each prediction model for Defects 1-3 with training and test data. The mode and mean of error histograms are on the order of 0.001-0.01 and both training and test data show nearly identical distribution. Defects 1-2 have larger standard deviation, which is consistent with the higher NRMSE at the edges of output spectral range and shorter wavelengths around 300 nm in Fig. 6, respectively. The largest kurtosis of Defect 1 for test data indicates tails of the distributions are heavy compared to normal distribution, mostly from the radiances of saturation pixels.

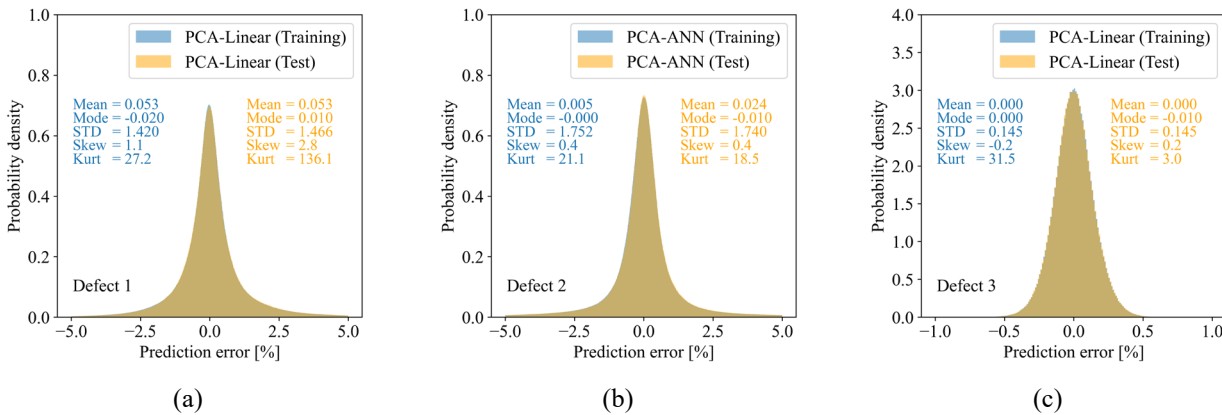

**Figure 7** Error histograms of randomly collected training (blue) and test (yellow) datasets measured in March 2021 with the optimized models for Defects 1-3 (PCA-ANN for Defect 2 and PCA-Linear for Defects 1 and 3). Prediction error and the statistics are calculated with the difference between the predicted and measured radiances divided by the latter.

## 3.2 Evaluation

### 3.2.1 Spatial inspection

For quantitative evaluation of the reproduced spectra, each defect area (Defects 1-3) and its surroundings where actual measurements regarded as 'true' exist are investigated. The evaluation is made with the data measured on 10 March 2021 (06 UTC), which are excluded for the model training. The center longitude of the areas is set to 128° E, which is identical to the sub-nadir longitude of GK-2B. Table 3 presents spectral ranges of Defects 1-3 and the target wavelengths for the analysis. Targeting the wavelengths for the analysis helps to specifically analyze the spectral patterns of absorption lines of trace gases and cloud properties.

**Table 3** The spectral range of Defects 1-3 and target wavelengths for the analysis. The third column presents GEMS retrieval products of which each fitting window is overlapped with Defects 1-3.

| Defect | Target wavelength | GEMS Level 2 product | Optimized model |
|---|---|---|---|
| 1 (400-500 nm) | 432-450 nm | CHOCHO, $NO_2$ | PCA-Linear |
| 2 (300-400 nm) | 312-360 nm | $O_3$, HCHO, $SO_2$, $NO_2$, aerosol optical depth | PCA-ANN |
| 3 (484-491 nm) | 484-491 nm | Cloud, AEH | PCA-Linear |

The measured and reproduced radiances with machine learning methods are directly compared, which are hereafter referred to as GEMS radiances and ML radiances. In Figs. 8-10, each column shows GEMS, ML radiances and the difference while the first and second rows show the radiances at the representative wavelengths showing the smallest and the largest differences, respectively. Figure 8 shows the comparison results of the Defect 3 area, which presents the best performance compared to the Defect 1 and 2 areas. The difference in Fig. 8 is within the range of ± 0.5% because the spectral gap of Defect 3 is narrower than the counterparts of Defects 1-2. For Defect 3, there is no distinct scene dependence over the output wavelengths and the difference shows noise-like features originated from instrument artifacts. One thing to be noted is that the results presented here is calculated at the finer spectral grids of 0.1 nm before interpolating to the original spectral grids. After the interpolation, the difference especially at strong peaks in a spectrum could increases by 0.5% for Fig. 8b.

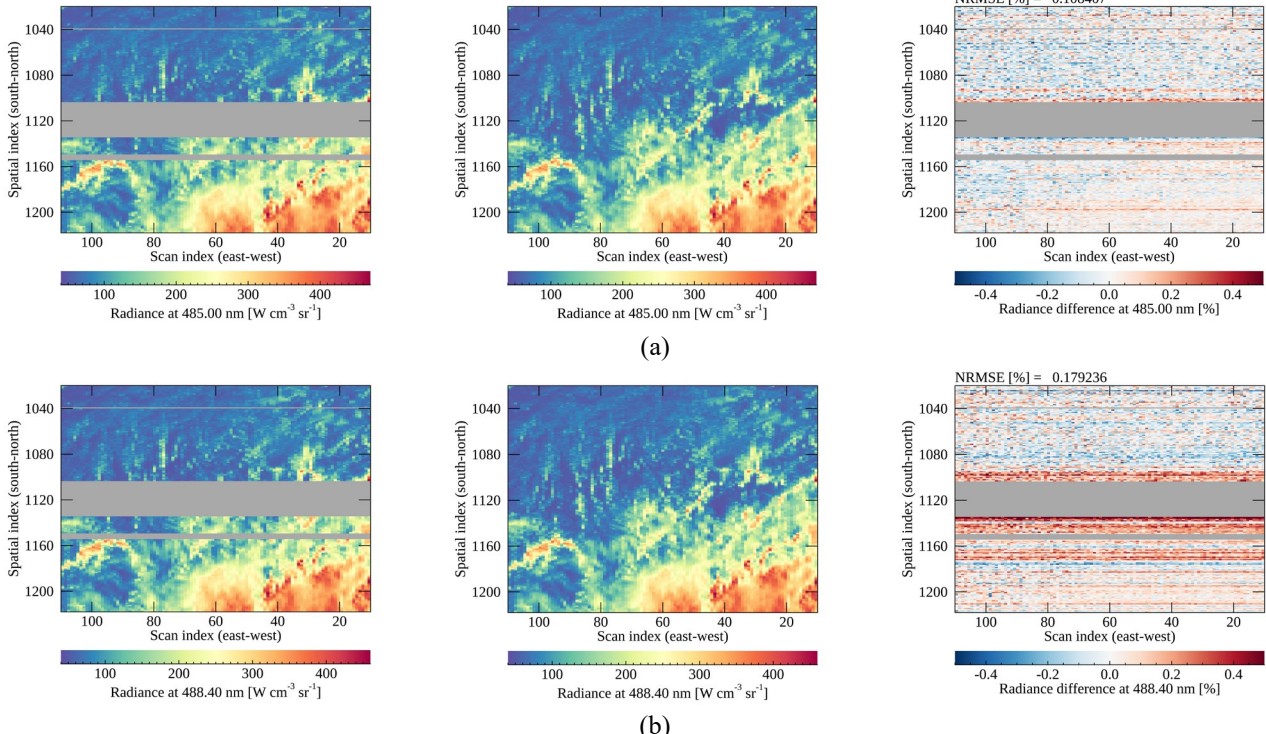

**Figure 8** Spatial distribution of GEMS, ML radiances and the difference (from the first to the third column) at the wavelengths presenting (a) the smallest and (b) the largest differences for the Defect 3 area. The difference is calculated between the ML and GEMS radiances divided by the latter in percent. Bad pixels are marked in dark gray and the color bar range is ± 0.5%. The unit of NRMSE is in percent divided by mean radiance.

Figure 9 shows the Defect 1 area where differences between GEMS and ML radiances are within about 5%. It shows that dark targets (clear sky with low radiance) show a positive difference while bright targets (mostly cloudy sky with high radiance) show the opposite. The tendency is also found on the other dates for different angle conditions. It seems the applied machine learning model (PCA-Linear) might have its limitation in describing the non-linear relations of angle conditions, scene properties and radiances causing the difference of about 5%.

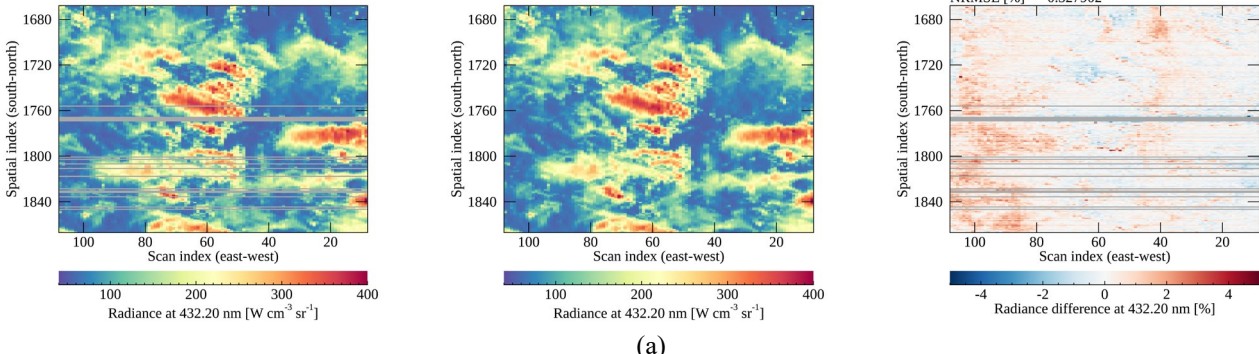

(a)

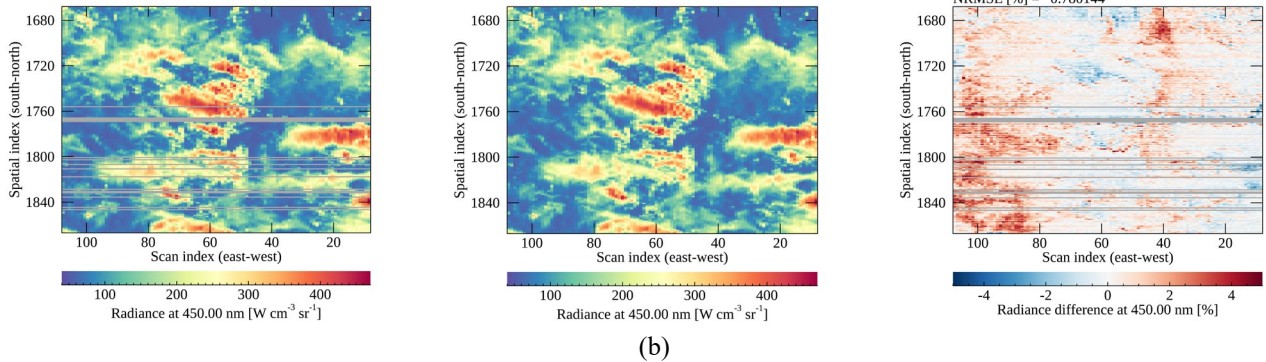

(b)

**Figure 9** Same as Fig. 8 for the Defect 1 area with the color bar range of 5%.

For the Defect 2 area, it is clear that the information from radiances of wavelengths longer than 400 nm is insufficient to effectively reproduce the spectral features at shorter wavelengths (consistent results with Figs. 6-7). Both Defects 2-3 have the output spectral ranges of about 100 nm but it seems the output radiances near 300 nm for Defect 2 need more information to be successfully reproduced. The stripping features found in Fig. 10b become significant at 312 nm for the ML radiances on the contrary to the radiances at 356.8 nm in Fig.10a. The stripping features seem to be added during the reproducing process especially for shorter wavelengths, and the reason is still unclear. We suspected that unpredictable noises from the instrument would cause the features and it seems more distinguishable in low signals. The scene dependence found in Fig. 9 is also dominant in Fig. 10 at shorter wavelengths, but with the opposite tendency. It is also found that some areas are undetected as bad pixels causing big differences over the areas close to the center in Fig. 10.

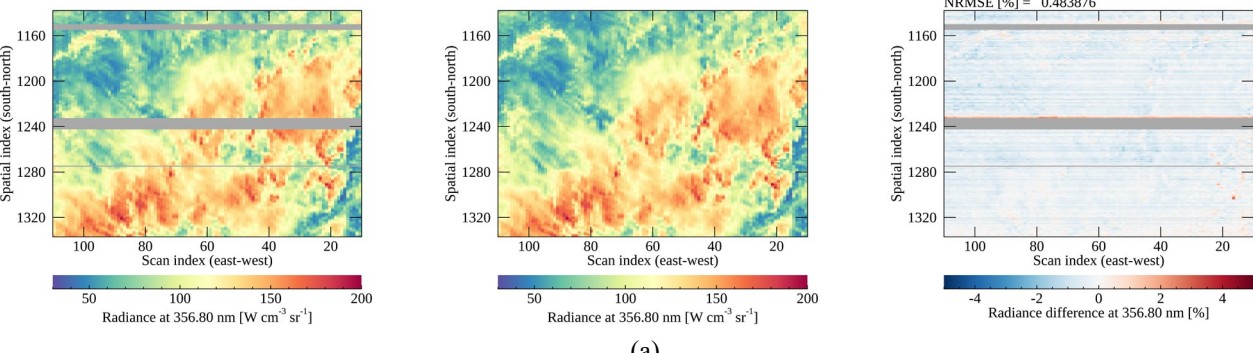

(a)

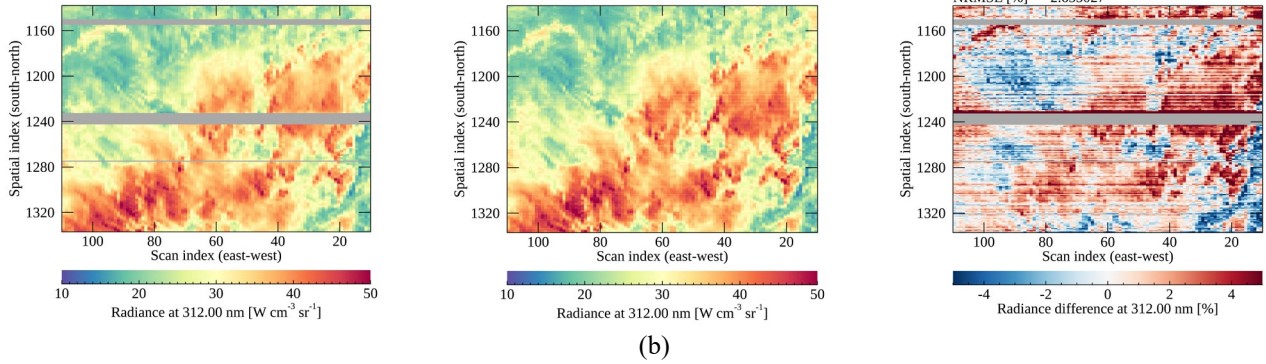

(b)

**Figure 10** Same as Fig. 8 for the Defect 2 area with the color bar range of 5%.

### 3.2.2 PCA-based analysis

275 To investigate further the reproduced spectral patterns before the retrieval process, we apply PCA to GEMS radiances collected within each area in Fig. 8-10 at the target wavelengths (see Table 3). With PCA, various spectral patterns are compressed to PC scores and this indicates that if a spectrum has disparate spectral patterns, the PC scores would also have distinct values when comparing with the PC scores of defect-free spectra. Figure 11 presents the PC scores of GEMS and ML radiances which are projected with the identical eigenvector matrix (corresponding to **X** in Eq. 1) constructed from GEMS radiances. The

280 Defect 3 area having the widest defective width is presented for the inspection and the second PC scores are used for the analysis because the first PC scores represent mean radiances as discussed in Sect. 3.1.1. The radiances reproduced with spatial interpolation on the bad pixel area are projected together as shown in Fig.11a, and it seems disparate values are found over the bad pixel area. The ML radiances in Fig. 11b show spatially homogenous PC scores on the contrary which indicates that the machine learning methods could reproduce dominant spectral patterns, in this case for the second PCs.

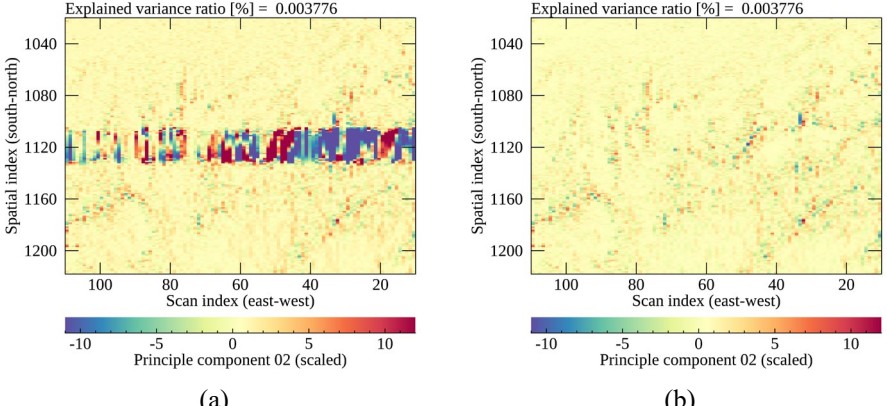

(a)          (b)

285 **Figure 11** The second PC scores of (a) GEMS radiances and (b) ML radiances on the target area for Defect 3. The PC is scaled for clarity of presentation.

The dominant spectral patterns for each PC are presented in Fig. 12 with the eigenvector matrix constructed from GEMS radiances for the specified target wavelengths in Table 3. Each color indicates the eigenvector for the first-sixth PCs contributing to total radiances at each wavelength. Li et al. (2015) verified that the leading PCs from the UV/VIS backscattered radiation (shorter than 360 nm) mainly represent dominant absorption features and surface properties, while the trailing PCs might be associated with instrument artifacts and other unresolved spectral features with PCA as similarly shown in Fig. 12.

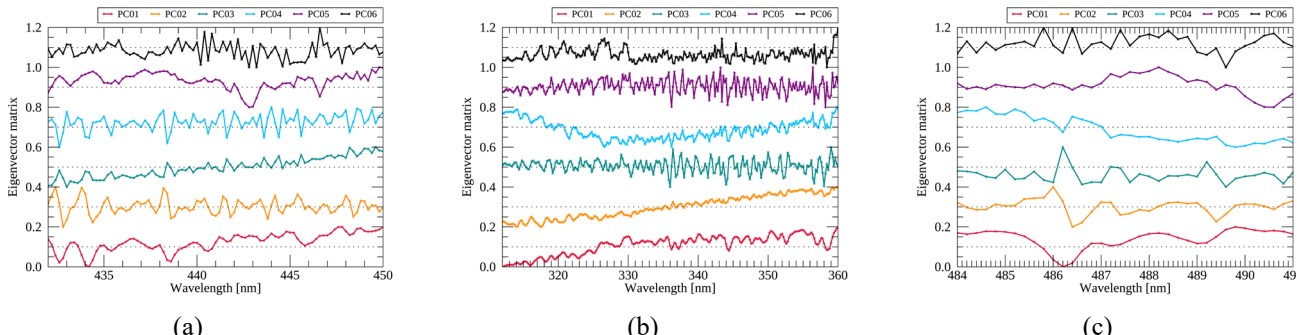

(a)                                    (b)                                    (c)

**Figure 12** Eigenvector of the first-sixth PCs applied to GEMS radiances for the target wavelengths of (a) Defects 1, (b) Defect 2 and (c) Defect 3. All eigenvectors are scaled (min-max scaling) and shifted for clarity of presentation.

As presented in Table 4, comparing PC scores provides qualitative information on the effectiveness of the suggested method. The results show that the mean spectral pattern (the first PC) and some dominant patterns can be sufficiently reproduced with the suggested models, but other spectral features such as the third PC for Defect 1 or the second PC for Defect 2 have difficulty obtaining valid information from input radiances for accurate reproduction. Given the explained variance ratio, each PC except the first one may contribute to a small extent to total radiances but it could be enough to determine subtle spectral patterns important for retrieval processes. The interesting finding is that only for Defect 3, the leading PCs having relatively higher explained variance ratio show high correlation coefficients over 0.95. The effectiveness of spectral replacement for each spectral region could be glimpsed in the results, which will be discussed further in the following section with retrieval process.

**Table 4** Correlation coefficients (Corr.) of PC scores of GEMS and ML radiances and explained variance ratios (EVR) of GEMS radiances for each target region in Fig. 8-10 excepting bad pixel area.

| PC | Defect 1 | | Defect 2 | | Defect 3 | |
|----|------|---------|------|---------|------|---------|
|    | Corr. | EVR | Corr. | EVR | Corr. | EVR |
| 1 | 0.9999 | 99.9906 | 0.9998 | 99.9504 | 1.0000 | 99.9953 |
| 2 | 0.9983 | 0.0070 | 0.8672 | 0.0294 | 0.9976 | 0.0038 |
| 3 | 0.8511 | 0.0007 | 0.9857 | 0.0135 | 0.9863 | 0.0003 |
| 4 | 0.9731 | 0.0006 | 0.5469 | 0.0019 | 0.8147 | 0.0001 |
| 5 | 0.6646 | 0.0001 | 0.8454 | 0.0012 | 0.6079 | 0.0001 |
| 6 | 0.7999 | 0.0001 | 0.7197 | 0.0005 | 0.7815 | 0.0001 |

### 3.3 Level 2 retrieval results

### 3.3.1 Cloud and ozone retrieval

In the previous section, it was found that the overall prediction error is about 5% except for ozone absorption lines and dominant spectral patterns can be successfully reproduced with the suggested method. The next question to be discussed is whether the reproduced spectral features are applicable to the retrieval process. Even if the trained models accurately reproduce an absolute value at each wavelength, the Level 2 retrieval could be unsuccessful if non-linear relations are too elusive to be properly emulated with the model. The radiances at $O_2$-$O_2$ absorption lines related to Defect 3 has the smallest error of 0.5% and we checked that cloud information with the fitting window in 460.2-490.0 nm can be successfully retrieved with the reproduced spectra presented in Fig. 8. The difference of cloud centroid pressure retrieved with ML and GEMS spectra is about 1% on average while the cloud properties retrieved with ML spectra have weak stripping features. The spectral range of Defect 3 is very narrow and thus the input radiances provide enough information for successful spectral replacement and the retrieval process.

For qualitative investigation of the replaced radiances at ozone absorption lines having high uncertainties, the reproduced spectra presented in Fig. 10 are applied to the ozone retrieval algorithm of GEMS. Figure 13 shows total ozone column density with un-flagged bad pixel area for the comparison of spatial discontinuity. As previously mentioned in Sect. 2.1.2, the ozone properties retrieved with measured GEMS spectra show distinct spatial discontinuity over the bad pixel area as shown in Fig. 13a and the discontinuity is somewhat reduced in Fig. 13b with ML spectra. However, the retrieved properties show different spatial distribution patterns even for the surrounding areas. It seems the ozone properties are underestimated especially for higher radiances in Fig. 13b and the stripping features found in Fig. 10 may affect the retrieval process considering the features are also found in Fig. 13b. It is also clear that the angle conditions provide important information for the retrieval because without the conditions, the retrieval results show unrealistic features with much higher variance. The results indicate that the spatial distribution could be approximated with reproduced spectra, but the accuracy could not be guaranteed especially for obtaining an exact retrieval value.

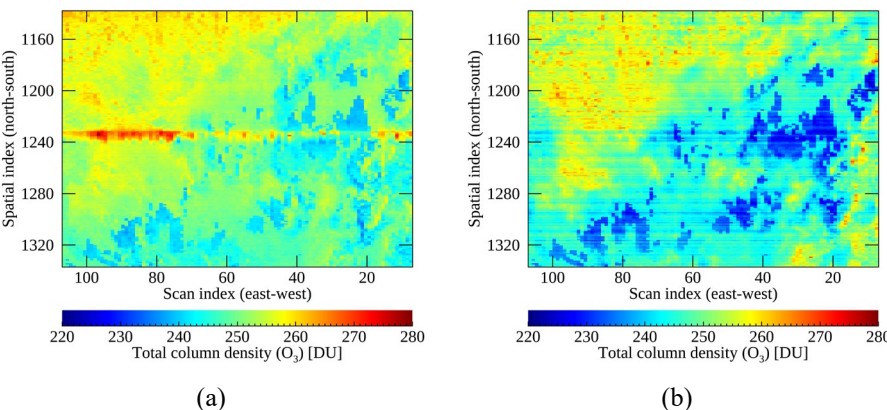

(a)                                               (b)

**Figure 13** Spatial distribution of total ozone column density retrieved with (a) GEMS and (b) ML radiances presented in Fig.11. The GEMS

spectra are measured on 10 March 2021 (06 UTC).

### 3.3.2 Cause analysis for further application

The retrieval uncertainty found in Fig. 13b is attributed to the lack of information in the input data or insufficient model optimization. For Defect 2, the input spectral range (400-500 nm) may have deficient information for ozone properties and it could cause the unsuccessful replacement. To clarify this and investigate further for future applicability, we choose two output

cases targeting ozone absorption lines in 312-360 nm and Fraunhofer lines in 390-400 nm to apply the suggested method with different input cases. In the Fraunhofer lines, the Ring effect caused by rotational Raman scattering can be found over two radiance peaks which is generally known to be very small and largely affected by the existence of clouds (Joiner et al., 1995). Together with ozone absorption lines, the analysis results could give a clear evidence on whether the small scattering features could be reproduced with machine learning depending on different input wavelengths. The PCA-ANN model is trained for

each input case respectively with defect-free measurements in March 2021 (around 80,000 spectra after bad pixel masking and the elimination of saturation pixels).

Figure 14 presents mean absolute errors for ozone absorption and Fraunhofer lines with different input conditions including or not the near sides of the output spectral bands (within the range of 20 nm). As assumed, prediction errors increase at the spectral peaks and overall error patterns along the output wavelengths largely differ depending on input conditions. It is

clearly shown that the errors are higher when reproduced with farther input spectral bands from output spectral lines. In Fig. 14a, the similar input condition (360-500 nm) with Defect 2 is plotted in black lines and the results clarify that the insufficient information from the input data may cause large errors for radiances at shorter wavelengths and subsequently the ozone retrieval process. Figure 14 verifies that each input case has a different amount of information determining the accuracy of the model to reproduce certain spectral features. It also can be deduced that the method could be quite useful even for strong

absorption lines when the input and output spectral ranges are sufficiently close.

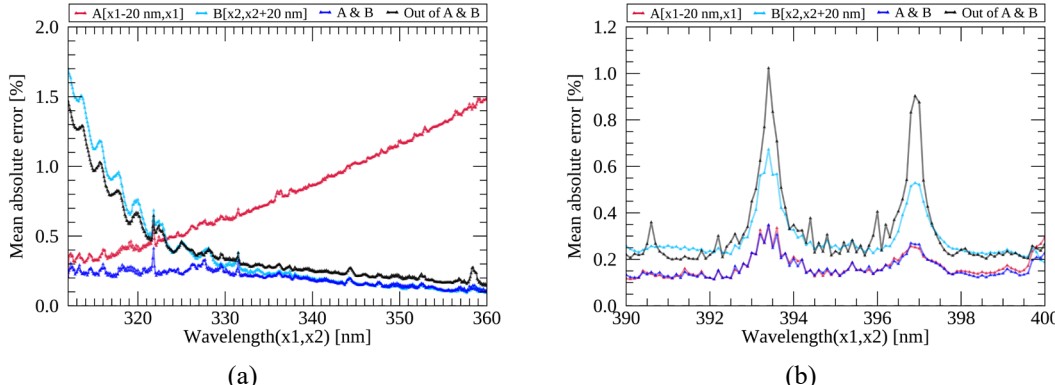

(a)                                                                              (b)

**Figure 14** Mean absolute errors for the reproduced and measured radiances at (a) ozone absorption and (b) Fraunhofer lines and the x1 and x2 indicates wavelengths at the edges of output spectral bands. The absolute error is calculated between the ML and GEMS radiances divided by the latter in percent.

Figure 15 presents a closer inspection by dividing spectra into four groups depending on the scene brightness. Different scenes could have different error levels which could be ignored in the averaged values in Fig. 14. The analysis is performed with the spectra reproduced with the input conditions showing the smallest (blue lines) and the largest (black lines) errors in Fig. 14. The PCA-ANN model reproduces dominant spectral features with an error of 0.4% for all scenes with the best input condition including near sides of output spectral bands as shown in Fig. 15, but it seems the difference increases

with darker scenes (weak signals). This indicates low signals would be generally less predictable even with the information extracted from the very close wavelengths. The error spectra show more distinguishable spectral features with farther input spectral bands, which shows that the spectral information from the input condition would be insufficient to properly reproduce exact spectral features.

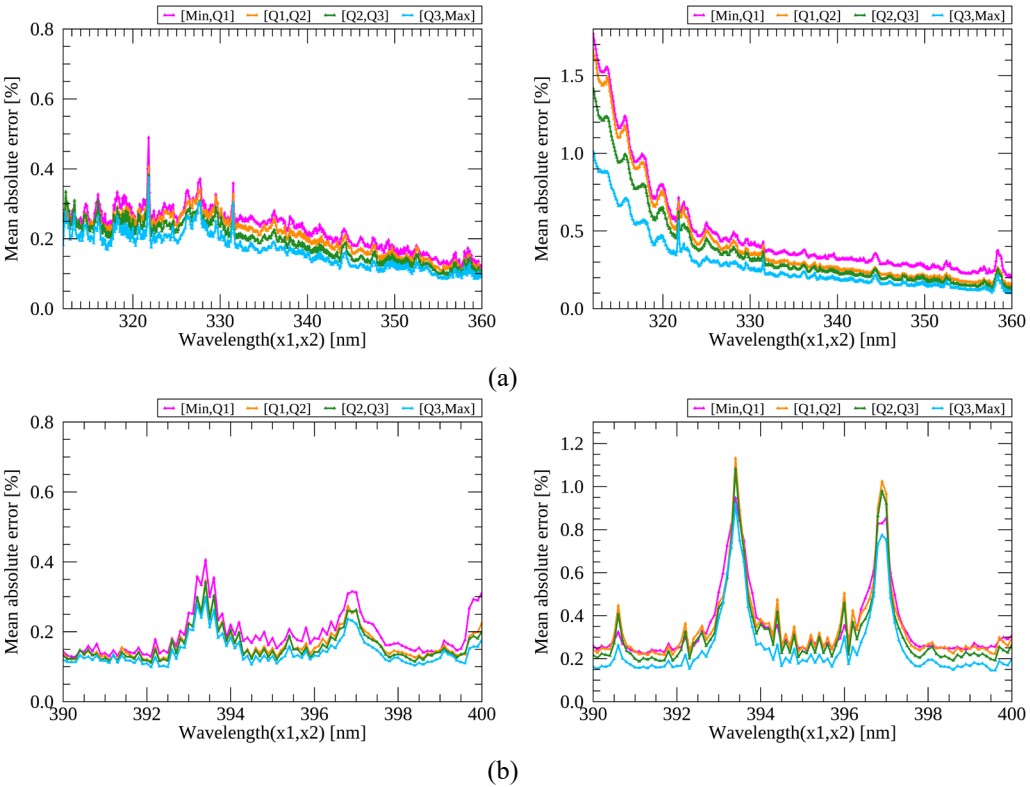

**Figure 15** Mean absolute errors for the reproduced and measured radiances at (a) ozone absorption and (b) Fraunhofer lines with different
input spectral bands including (the first column) or excluding (the second column) near sides of output wavelengths within the range of 20-nm. The Q1, Q2 and Q3 represent the first, second and third quartile and each color indicates the average in the range of each quartile. The x1 and x2 indicates wavelengths at the edges of output spectral bands and the absolute error is calculated between the ML and GEMS radiances divided by the latter in percent.

        In this section, different output spectral bands containing absorption or scattering lines are compared with different
input conditions. It seems the suggested method (PCA-ANN) could be quite effective when the input spectral ranges are closer to the target wavelengths to be reproduced. However, it is not necessarily true the wider the input spectral range is, the more accurate the replacement becomes. If input spectral ranges have some calibration issues (e.g. stray light or saturation) or

provide conflicting features with other input spectral bands as shown in Fig. 14a, the reproduced spectrum would have diverse and inconsistent features causing higher error. In conclusion, the suggested method accurately predicts the overall magnitude

of a spectrum, but reproducing a certain spectral feature with high accuracy would need more information especially for low signals or strong absorption lines. At least, the input and output spectral regions should be close enough to reduce the spectral error up to 0.5%, the uncertainty of the reproduced spectra at $O_2$-$O_2$ absorption lines presenting successful cloud retrieval results.

## 4 Conclusions

GEMS is an environmental sensor measuring hyperspectral radiances from 300 to 500 nm in the Asia-Pacific region for timely atmospheric monitoring. During the IOT of GEMS, one of calibration issues was found that erroneous values of bad pixels on the detector array are not properly replaced with spatial interpolation, the current operational method of GEMS. It is clear that when the bad pixel area is too large, the spatial interpolation tends to cause high interpolation error especially for a scene having large spatial inhomogeneity (i.e. cloud edges). The high interpolation error of bad pixels could affect the retrieval

process, which causes horizontal discontinuity at a certain latitude for the retrieval of Level 2 products.

In terms of accuracy, the spatial gaps found in Level 2 products could be better improved when applying a fitted method based on spatial distribution characteristics of each product. In this regard, we more focus on improving the erroneous spectrum itself on the radiance level to check whether the issue could be more efficiently resolved for both radiances and retrieved properties with improved spectral features. For the approach, this study suggests machine learning methods (PCA-

ANN and PCA-Linear) to fill in various spectral gaps denoted as Defects 1-3 by investigating how much information could be obtained to reproduce spectral features without any additional information. The basic assumption of this approach is that radiances of a spectrum have strong linear and non-linear relations, which could be emulated with the ANN and multivariate linear regression. The spectral range of output radiances is set to the wavelengths of bad pixels, while the input radiances correspond to the rest part of a spectrum for Defects 1-3, respectively.

In the results, PCA-Linear model presents smaller prediction errors for the defective region having strong linear relations between input and output radiances (Defect 1) or having a narrower spectral gap (Defect 3). When applying the reproduced spectra for Defect 3 to the cloud retrieval, the cloud centroid pressure is successfully retrieved with an error of 1%, on average. This is because the output spectral range of Defect 3 is comparably narrower and thus the input wavelengths provide enough information to reproduce exact spectral features which are valid for the subsequent retrieval process. The PCA-

ANN model is better for the output radiances having strong non-linear relations (Defect 2). Dominant spectral patterns and the overall magnitude of spectra could be successfully reproduced mostly with an error of 5% except for ozone absorption lines, while the exact spectral patterns would be insufficiently reproduced. When applying the reproduced spectra to the ozone retrieval, the spatial distribution of total ozone column density can be approximated but with high uncertainty. This indicates

that additional information would be needed except for the spectral relations of radiances for the successful retrieval process at strong absorption lines ultimately reducing spatial gaps in the Level 2 products.

Considering that the number of bad pixels would increase in operation as did in Ozone Mapping and Profiler Suite (OMPS) (Seftor et al., 2014), an efficient way of replacing bad pixels would be necessary for the long-term operation of GEMS. It is also highly possible that an unexpected issue could occur such as the row-anomaly of Ozone Monitoring Instrument (OMI) (Schenkeveld et al., 2017). The ultimate goal of this research is to increase the usefulness of GEMS data for a longer time period, at least for designed lifetime of ten years. The current work verifies that the gap filling (in Level 1) with certain spectral conditions shows quite reliable results even with the limitations for the strong absorption bands, which is natural and provides the reasons why we need observation data over such spectral bands. However, it is also anticipated that accumulation of observation data along with auxiliary data and improved nonlinear algorithm, the limitation could be improved in future study. For that, this paper provides the basis for further applicability of the method by evaluating the efficiency of machine learning methods to reproduce hyperspectral data especially in the UV/VIS spectral range.

**Author contribution**

M.-H.A. conceptualized and supervised the study; Y.L. conducted the research, performed the experiments and prepared the manuscript; M.K. contributed to the editing of the manuscript and developing methodology. M.E. contributed to the pre-processing of raw data.

**Competing interests**

The authors declare that they have no conflict of interest.

**Acknowledgements**

We wish to express our gratitude to Dr. Glen Jaross and the anonymous reviewer for their valuable comments to greatly improve the quality of this research. We also thank Dr. Kang-Hyeon Baek (Pusan National University) and Gyuyeon Kim (Ewha Womans University) for their assistance in retrieving Level 2 data for this study. The authors acknowledge the contribution of the Environment Satellite Center (ESC) of National Institute of Environmental Research (NIER) for providing GEMS Level 0-1C data.

**Financial support**

This research was supported by Basic Science Research Program through the National Research Foundation of Korea(NRF) funded by the Ministry of Education(2018R1A6A1A08025520).

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
