# Peer review of "Spectral replacement using machine learning methods for continuous mapping of Geostationary Environment Monitoring Spectrometer (GEMS)"

_Atmospheric Measurement Techniques, 2022_

## Author Comment (AC1)

**Manuscript ID**: amt-2022-37

**Spectral replacement using machine learning methods for continuous mapping of Geostationary Environment Monitoring Spectrometer (GEMS)**

Yeeun Lee, Myoung-Hwan Ahn*, Mina Kang, Mijin Eo

**Reply to comments #2 from Dr. Glen Jaross**

We would like to thank Glen Jaross for his insightful comments and suggestions, which are significant to reconsider the direction of our research and include some missed points not fully explained in the first draft. We have revised the manuscript considering the suggested comments and in this document, we will go over major concerns raised by the referee and describe how to have improved the manuscript accordingly. The referee's comments are presented in **bold** and the presented figures and tables have been included in the revised manuscript with identical numbering. The revised manuscript is presented in *italic*.

**\<General comments\>**

**Comment – Part #1:**

**"I have difficulty understanding the goals of this paper. There seems to be a mismatch between what the authors are trying to achieve and the investigation they describe in the paper. The authors describe a problem on GEMS whereby "bad" pixels, i.e. missing pixel radiances, cause problems in the ensemble of measured radiances. … There is no mention of what criteria are used to assess the results.**

**Response #1:**

We agree with the comments that the goals of this paper are not fully explained. Bad pixels of GEMS could cause distinct spatial discrepancy in GEMS Level 1 data and it definitely introduces errors in Level 2 products (or make impossible to derive the Level 2 products). Thus the primary goal of the current work is to evaluate the applicability of machine learning methods for the spectral gap filling, which could reduce bad pixel effects to both Level 1 and 2 products. For the purpose, bad pixel effects are discussed first (as suggested by the referee) and thus we have added the following part to Section 2.1, describing the effects of bad pixels to radiances and retrieved properties (i.e., ozone):

*The interpolation error could seriously affect the Level 2 products of which the spectral fitting windows are overlapped with bad pixel areas. For instance, cloud properties and aerosol effective height (AEH) of GEMS are retrieved from $O_2$-$O_2$ absorption bands around 477 nm (Choi et al., 2021; Kim et al., 2021) where the cluster of bad pixels is located (Defect 3). During the IOT, Defect 3 caused spatial discontinuity to the retrieved cloud and AEH distribution, which made the fitting window of the products modified to avoid bad pixel effects. Ozone retrieval is also affected by Defect 2 (300-400 nm) as the spectral radiances within 300-380 nm provide major information for the ozone retrieval of GEMS (Bak et al., 2019). When specifying Defect 2 and its surrounding pixels, the bad pixel effects are clearly*

*shown in Fig. 2 which presents GEMS radiances at 312 nm and the retrieved ozone total column density. Even though radiances at a certain wavelength are homogeneous with its surroundings (see Fig. 2a), the spectral patterns are not properly reproduced with the existed method (spatial interpolation) causing the retrieval errors easily found in the Level 2 product. The detailed discussion of reproducing spectral patterns of bad pixels will be continued in Sect. 3.2.2 with PCA-based evaluation.*

[Figure]

(a)                                                     (b)

**Figure 1** *Spatial distribution of (a) the GEMS measured radiance at 312 nm and (b) ozone total column density by zooming in the bad pixel area for Defect 2 and its surrounding area.*

**Comment – Part #2:**

**"From an academic perspective the question of how well ML techniques can describe Earth backscattered radiances is an interesting one. If the authors approached this paper from that perspective they might provide a useful contribution to our ability to characterize the atmosphere through numerical techniques. But in doing so they must take a more rigorous approach to evaluating their radiance predictions.**

**The first thing the authors should do is to forget about the GEMS pixel defects. These are of no use in evaluating the efficacy of the technique since the true radiances remain unknown for these regions. Instead, choose regions of the detector where there are good measurements and treat them as missing for the purpose of deriving errors. There can be a variety of region shapes and sizes, including ones that look very similar to Defects 1, 2, and 3."**

**Response 2:**

The points are also raised by another referee and we appreciate the suggestions. Following the suggestions, we targeted a certain area including each defect (Defects 1-3) and its surroundings (100-indices toward both north and south directions) where actual measurements (regarded as 'true') could be obtained. The results are added to the 'Results and discussion' (Section 3) which has been re-organized with 'Model selection' (Section 3.1) and 'Evaluation' (Section 3.2). The following part has been inserted to Section 3.2:

**3.2 Evaluation**

**3.2.1 Spatial and spectral inspection**

For the quantitative evaluation of the reproduced spectra, certain areas are targeted which include each defect (Defects 1-3) and its surroundings where actual measurements regarded as 'true' could be obtained. The evaluation is made with the data measured on 10 March 2021 (06 UTC), which are not used for the model training. The center longitude of the areas is set to 128° E, which is identical with the sub-nadir longitude of GK-2B. Along the spectral direction, we focus on the specific spectral range of the whole spectral gap of Defects 1-3, as shown in Table 3. Specifying the range helps to closely analyze the spectral patterns of absorption lines of trace gases and cloud properties. Table 3 presents spectral ranges of Defects 1-3 and the target wavelengths for the analysis.

*Table 1 The spectral range of Defects 1-3 and target wavelengths for the analysis. The third column presents GEMS retrieval products of which each fitting window is overlapped with Defects 1-3.*

| Defect | Target wavelength | GEMS Level 2 product | Optimized model |
|---|---|---|---|
| 1 (400-500 nm) | 432-450 nm | CHOCHO, NO2 | PCA-Linear |
| 2 (300-400 nm) | 312-360 nm | O3, HCHO, SO2, NO2, aerosol optical depth | PCA-ANN |
| 3 (484-491 nm) | 484-491 nm | Cloud, AEH | PCA-Linear |

For the evaluation, actual GEMS radiances and the reproduced radiances with machine learning methods are directly compared, hereafter called GEMS radiances and ML radiances, respectively. In Figs. 11-13, each column shows GEMS, ML radiances and the difference while the first and second rows show the representative wavelengths for the smallest and the largest difference, respectively. Figure 11 shows the comparison results of the Defect 3 area, which shows the best performance compared to the Defects 1-2 areas. The difference in Fig. 11 is close to zero (within ± 0.5%) because the spectral gap of Defect 3 is narrower than the counterparts of Defects 1-2. The narrower the spectral range of the output radiances is, the more abundant information could be obtained from the input radiances. For Defect 3, there is no scene dependence over the output wavelengths and the difference shows noise-like features except for the spatial dependence which might be originated from instrument artifacts.

[Figure]

*(a)*

[Figure]

*(b)*

**Figure 2** *The GEMS, ML radiances and the difference from left to right at the wavelengths presenting (a) the smallest and (b) the largest difference for the Defect 3 area. Bad pixels are marked in dark gray and the difference is calculated as (ML-GEMS)/GEMS in percent. The color bar range for the difference is $\pm$ 0.5% and the unit of RMSE is in percent divided by the mean radiance.*

Figure 12 shows the Defect 1 area where the ML radiances are within about 5% of the GEMS radiances. It also shows that dark targets (clear sky with small radiance) show a positive difference while bright targets (mostly cloudy sky with large radiances) show an opposite tendency. The tendencies are also found from the ML radiances on the other dates for different angle conditions such as SZA and VZA. It seems the applied machine learning model (PCA-Linear) might not be fully trained to resolve the different atmospheric conditions and radiances which causes a certain bias depending on the scenes.

[Figure]

*(a)*

*(b)*

**Figure 3** *Same as Fig. 11 for the Defect 1 area.*

For the Defect 2 area, it is clear that the information from valid radiances of wavelengths longer than 400 nm is insufficient to effectively reproduce the spectral features at shorter wavelengths (consistent results with Figs. 8-9). Both output spectral lengths of Defects 2-3 are nearly identical around 100 nm but it seems radiances near 300 nm need more information to be successfully reproduced.

The stripping feature found in Fig. 12b is significant at 312 nm for the ML radiances, while it doesn't at 357.2 nm in Fig.12a. The stripping feature seems to be added during the reproducing process especially for shorter wavelengths, and the reason is still unclear. Another distinct feature found in Fig. 12 is that the difference in northern parts is very large with the difference of 10%. We suspect that the reason might be the VZA effect considering that VZA increases at the northern parts in the area. Without angle conditions in the input parameters for the model, the difference becomes doubled at 312 nm presenting similar patterns with the difference in Fig. 12b. This indicates the angle effect can be emulated in the model by applying VZA and SZA as the input parameters, but it is not fully resolved especially for the radiances at shorter wavelengths.

[Figure]

*Figure 4* Same as Fig. 11 for the Defect 2 area.

A closer inspection is performed to analyze the general spectral features over target wavelengths. For each defect area in Figs. 11-13, the collected spectra are divided into four groups depending on the scene brightness considering that ML radiances could have different systematic biases depending on the scenes. With the data, the mean difference is calculated for each wavelength. As found in Fig. 11, Fig. 14a shows that the ML radiances over dark scenes have the positive bias while brighter scenes have the negative bias. It is interesting that the scene dependence is only significantly found for Defect 1. Figure 14b indicates that the ML radiances are overestimated except for the very brighter scenes. It should be noted that the y-axis range of Fig.14b is wider than the figures for Defects 1 and 3. With the results, it can be deduced that the complicated atmospheric effects at the shorter wavelengths are difficult to be emulated and instrument artifacts such as stray light also would affect the reproducing process. Figure 14c shows relatively large difference at the spectral peaks, but generally the difference is smaller than 0.2%

[Figure]

(a)     (b)     (c)

**Figure 5** *Mean difference between ML and GEMS radiances within the target area of (a) Defect 1, (b) Defect 2 and (c) Defect 3. Each color indicates the average for each quartile and Q1, Q2 and Q3 represent the first, second and third quartile, respectively. The difference is calculated as (ML-GEMS)/GEMS in percent.*

Besides the shorter wavelengths of Defect 2, mean ML radiance and the difference with GEMS radiances are presented by targeting Fraunhofer lines from 390 to 400 nm (see Fig. 14). The Ring effect caused by rotational Raman scattering can be found over the two peaks in Fig. 14a, which is generally known to be very small and largely affected by clouds (Joiner et al., 1995). Figure 14b shows that PCA-ANN reproduces the dominant features at the peaks very well on average within 0.6%, but it seems the difference increases with darker scenes where the Ring effect becomes stronger. This indicates that the ML radiances would need additional information to successfully reproduce the exact spectral features especially for the very small signals such as the Ring effect.

[Figure]

(a)     (b)

**Figure 6** *(a) Mean ML radiances (b) and the difference with GEMS raidances at the Fraunhofer lines for the Defect 2 area. Each color indicates the average for each quartile and Q1, Q2 and Q3 represent the first, second and third quartile, respectively. The difference is calculated as (ML-GEMS)/GEMS in percent.*

**3.2.2 PCA-based spectral analysis**

*As applied in the pre-processing step in our research, PCA is a very useful tool to capture the meaningful variances along the spectral direction and it has been widely used to retrieve environmental and surface properties (Horler and Ahern, 1986; Joiner et al., 2016; Li et al., 2013, 2015). To investigate further the spectral patterns, we apply PCA to GEMS radiances (except for bad pixels) at the target wavelengths (see Table 3) collected within each area in Fig. 11-13. With PCA, various spectral patterns are compressed and a spectrum can be projected to PC subspaces by multiplying with the constructed PC matrix (eigenvector matrix). This indicates that if a spectrum has disparate spectral patterns, the projected PCs would also have distinct values when comparing with the PCs of GEMS radiances. Figure 15 presents the results when projecting both GEMS and ML radiances with PCA. For the inspection, the Defect 3 area is presented which has the wider defective width along the north-south*

direction. Because the first PC scores represents mean radiances, the second PC are used for the analysis. As we assumed, bad pixels in Fig. 15a show disparate values because the spectral patterns of the interpolated spectra are inconsistent with GEMS radiances. The ML radiances in Fig. 15b show spatially homogenous PC scores which indicates that the machine learning methods could properly reproduce the dominant spectral patterns, in this case of the second PC.

[Figure]

(a)                                        (b)

**Figure 15.** *The second PC of (a) actual measurements and (b) reproduced spectra on the target area for Defect 3. The PC is scaled for clarity of presentation.*

The dominant patterns for each PC are presented in Fig. 16 with GEMS radiances for the target wavelengths of Defects 1-3. Each color indicates the eigenvector of the first-sixth PCs which determines how each PC score of a spectrum contributes to the original spectrum. Li et al. (2015) verified that the leading PC scores from the UV/VIS backscattered radiation (shorter than 360 nm) are significantly correlated with dominant absorption features and surface properties. The trailing PC scores might be associated with instrument artifacts and other unresolved spectral features. Figure 16 shows that the first PC corresponds to the mean spectrum as discussed in Sect. 3.1.1 and the second-sixth PCs show dominant spectral patterns originated from absorption features of trace gases, surface properties and unresolved features. This indicates that the comparison of PC scores could provide the information on the similarity of the dominant patterns between ML and GEMS radiances as shown in Table 4 with the correlation coefficient. The results indicate that the mean spectral feature (the first PC) and some dominant patterns (the second and third PCs) could be well reproduced with the suggested models, but other spectral features such as the fourth PC for Defect 2 have difficulty obtaining valid information from input radiances for accurate reproduction. The magnitude of radiance from the major PCs except for the first PC might not be large considering that even the leading PCs have small explained variance ratio for hyperspectral data in UV/VIS spectrum. However, it would be enough to determine the exact spectral signals which are mostly related to the important information for the retrieval process.

[Figure]

**Figure 7** *Eigenvector of the first-sixth PCs applied to GEMS radiances for the target wavelengths of (a) Defects 1, (b) Defect 2 and (c) Defect 3. All eigenvectors are scaled (min-max scaling) and shifted for clarity of presentation.*

**Table 2.** *Correlation coefficient of PC scores of reproduced and actual measurements for Defects 1-3.*

| Defects | PC 01 | PC 02 | PC 03 | PC 04 | PC 05 | PC 06 |
|---------|-------|-------|-------|-------|-------|-------|
| Defect 1 | 0.9999 | 0.9976 | 0.8172 | 0.9779 | 0.6846 | 0.6609 |
| Defect 2 | 0.9999 | 0.8129 | 0.9876 | 0.4294 | 0.7035 | 0.5046 |
| Defect 3 | 0.9999 | 0.9962 | 0.9787 | 0.6644 | 0.5399 | 0.2649 |

**Comment – Part #3:**

**"But what is the real problem that the authors are trying to solve? In an instrument such as GEMS, designed to measure trace gas composition of the atmosphere through hyperspectral measurements, the goal is probably to produce trace gas products without spatial gaps caused by missing radiances. The authors do not discuss the issue of trace gas retrievals or other products derived from their predicted radiances. Is there any improvement at all in those products? …**

**The authors must also devise evaluation criteria that are more robust and quantitative than "these spectra look realistic." Since the goal for GEMS radiances is to derive atmospheric products such as trace gases, perhaps these trace gas retrievals can be used as the metric. Merely stating that predicted radiances agree on average with measured radiances to within X% ignores the subtle spectroscopic sensitivity of trace gases such as NO2, where the exact relationship between wavelengths is of utmost importance."**

**Response 3:**

As presented in the previous section, we have evaluated the reproduced spectra by comparing with actual measurements and applied PCA to analyze spectral features of reproduced spectra. With the analysis, it was found that the machine learning models properly reproduce dominant spectral patterns for Defects 1-3 with only radiances from the rest part of spectra and angle conditions. However, the exact spectral features (<1%) determined by small signal (the important information) may be accurately reproduced only if the spectral range of output radiances are closer to the input radiances enough to obtain sufficient information from the input radiances (such as Defect 3). Also, it seems additional information would be needed to reproduce exact spectral features when the output radiances are overlapped with strong absorption or scattering lines. Considering the ultimate goal of measuring hyperspectral data, we agree with the referee's suggestion to apply the retrieval algorithms for evaluating the reproduced spectra. However, as the initial approach reproducing missing radiance of GEMS, we hope to evaluate the applicability of machine learning methods for the GEMS measurements, which have meaningful information as well as instrument artifacts. As the referee pointed out, the effect of reproducing spectra for the retrieval process is a necessary step and based on the findings in this research, we hope to investigate further the step in a follow-up study.

**Reference**

Bak, J., Baek, K. H., Kim, J. H., Liu, X., Kim, J. and Chance, K.: Cross-evaluation of GEMS tropospheric ozone retrieval performance using OMI data and the use of an ozonesonde dataset over East Asia for validation, Atmos. Meas. Tech., 12(9), 5201–5215, doi:10.5194/amt-12-5201-2019, 2019.

Choi, H., Liu, X., Gonzalez Abad, G., Seo, J., Lee, K.-M. and Kim, J.: A Fast Retrieval of Cloud Parameters Using a Triplet of Wavelengths of Oxygen Dimer Band around 477 nm, Remote Sens., 13(1), 152, doi:10.3390/rs13010152, 2021.

Horler, D. N. and Ahern, F. J.: Forestry information content of thematic mapper data, Int. J. Remote

Sens., 7(3), 405–428, doi:10.1080/01431168608954695, 1986.

Joiner, J., Bhartia, P. K., Cebula, R. P., Hilsenrath, E., McPeters, R. D. and Park, H.: Rotational Raman scattering (Ring effect) in satellite backscatter ultraviolet measurements, Appl. Opt., 34(21), 4513, doi:10.1364/AO.34.004513, 1995.

Joiner, J., Yoshida, Y., Guanter, L. and Middleton, E. M.: New methods for the retrieval of chlorophyll red fluorescence from hyperspectral satellite instruments: Simulations and application to GOME-2 and SCIAMACHY, Atmos. Meas. Tech., 9(8), 3939–3967, doi:10.5194/amt-9-3939-2016, 2016.

Kim, G., Choi, Y. S., Park, S. S. and Kim, J.: Effect of solar zenith angle on satellite cloud retrievals based on O2–O2 absorption band, Int. J. Remote Sens., 42(11), 4224–4240, doi:10.1080/01431161.2021.1890267, 2021.

Li, C., Joiner, J., Krotkov, N. A. and Bhartia, P. K.: A fast and sensitive new satellite SO2 retrieval algorithm based on principal component analysis: Application to the ozone monitoring instrument, Geophys. Res. Lett., 40(23), 6314–6318, doi:10.1002/2013GL058134, 2013.

Li, C., Joiner, J., Krotkov, N. A. and Dunlap, L.: A new method for global retrievals of HCHO total columns from the Suomi National Polar-orbiting Partnership Ozone Mapping and Profiler Suite, Geophys. Res. Lett., 42(7), 2515–2522, doi:10.1002/2015GL063204, 2015.

---

## Author Comment (AC2)

**Manuscript ID**: amt-2022-37

**Spectral replacement using machine learning methods for continuous mapping of Geostationary Environment Monitoring Spectrometer (GEMS)**

Yeeun Lee, Myoung-Hwan Ahn*, Mina Kang, Mijin Eo

**Reply to comments #2 from the anonymous referee**

We would like to thank in advance the anonymous referee for the thorough reading and the detailed comments on the manuscript, which are very helpful to improve insufficient parts in the first draft. The manuscript has been revised following the comments below and we hope that all the comments have been addressed accordingly. The referee's comments are presented in **bold** and we apply the figure and table numbering of the revised manuscript. The revised manuscript is presented in *italic*.

**<General comments>**

**Comment #1:**

**"I believe the demonstration of the method validity can be improved. In particular, the qualitative discussion about the performance of reproduction, described with Figs. 11-12, needs to be improved and more objective. Also, in addition to the prediction errors presented in Figs. 7–9, it is recommended to show how well the proposed methods reproduce known good spectra (i.e., actual measurements)."**

**Response #1:**

Firstly, thanks for the valuable comments and suggestions. As the referee pointed out, we acknowledged that the result parts referred in the comments definitely need improvements and thus the Sections 2-3 have been greatly revised. The 'Results and discussion' part has been re-organized with 'Model selection' (Section 3.1) and 'Validation' (Section 3.2). The details will be discussed in the following section. Sections 3.1-3.2 in the first draft have been combined to Section 3.1.

The applied analysis in the first draft could not quantitatively prove the validity of the suggested methods. Following the two referee's recommendations, we targeted a certain area including each defective region (Defects 1-3) and its surroundings (100-indices toward both north and south direction) where actual measurements (regarded as 'true') could be obtained. The following part has been inserted to Section 3.2.

*3.2 Evaluation*

*3.2.1 Spatial and spectral inspection*

*For the quantitative evaluation of the reproduced spectra, certain areas are targeted which include each defect (Defects 1-3) and its surroundings where actual measurements regarded as 'true' could be obtained. The evaluation is made with the data measured on 10 March 2021 (06 UTC), which are not used for the model training. The center longitude of the areas is set to 128° E, which is identical with*

the sub-nadir longitude of GK-2B. Along the spectral direction, we focus on the specific spectral range of the whole spectral gap of Defects 1-3, as shown in Table 3. Specifying the range helps to closely analyze the spectral patterns of absorption lines of trace gases and cloud properties. Table 3 presents spectral ranges of Defects 1-3 and the target wavelengths for the analysis.

**Table 1** *The spectral range of Defects 1-3 and target wavelengths for the analysis. The third column presents GEMS retrieval products of which each fitting window is overlapped with Defects 1-3.*

| Defect | Target wavelength | GEMS Level 2 product | Optimized model |
|---|---|---|---|
| 1 (400-500 nm) | 432-450 nm | CHOCHO, NO2 | PCA-Linear |
| 2 (300-400 nm) | 312-360 nm | O3, HCHO, SO2, NO2, aerosol optical depth | PCA-ANN |
| 3 (484-491 nm) | 484-491 nm | Cloud, AEH | PCA-Linear |

For the evaluation, actual GEMS radiances and the reproduced radiances with machine learning methods are directly compared, hereafter called GEMS radiances and ML radiances, respectively. In Figs. 11-13, each column shows GEMS, ML radiances and the difference while the first and second rows show the representative wavelengths for the smallest and the largest difference, respectively. Figure 11 shows the comparison results of the Defect 3 area, which shows the best performance compared to the Defects 1-2 areas. The difference in Fig. 11 is close to zero (within $\pm$ 0.5%) because the spectral gap of Defect 3 is narrower than the counterparts of Defects 1-2. The narrower the spectral range of the output radiances is, the more abundant information could be obtained from the input radiances. For Defect 3, there is no scene dependence over the output wavelengths and the difference shows noise-like features except for the spatial dependence which might be originated from instrument artifacts.

[Figure]

**Figure 1** *The GEMS, ML radiances and the difference from left to right at the wavelengths presenting (a) the smallest and (b) the largest difference for the Defect 3 area. Bad pixels are marked in dark*

*gray and the difference is calculated as (ML-GEMS)/GEMS in percent. The color bar range for the difference is ± 0.5% and the unit of RMSE is in percent divided by the mean radiance.*

Figure 12 shows the Defect 1 area where the ML radiances are within about 5% of the GEMS radiances. It also shows that dark targets (clear sky with small radiance) show a positive difference while bright targets (mostly cloudy sky with large radiances) show an opposite tendency. The tendencies are also found from the ML radiances on the other dates for different angle conditions such as SZA and VZA. It seems the applied machine learning model (PCA-Linear) might not be fully trained to resolve the different atmospheric conditions and radiances which causes a certain bias depending on the scenes.

[Figure]

*(a)*

*(b)*

**Figure 2** *Same as Fig. 11 for the Defect 1 area.*

For the Defect 2 area, it is clear that the information from valid radiances of wavelengths longer than 400 nm is insufficient to effectively reproduce the spectral features at shorter wavelengths (consistent results with Figs. 8-9). Both output spectral lengths of Defects 2-3 are nearly identical around 100 nm but it seems radiances near 300 nm need more information to be successfully reproduced. The stripping feature found in Fig. 12b is significant at 312 nm for the ML radiances, while it doesn't at 357.2 nm in Fig.12a. The stripping feature seems to be added during the reproducing process especially for shorter wavelengths, and the reason is still unclear. Another distinct feature found in Fig. 12 is that the difference in northern parts is very large with the difference of 10%. We suspect that the reason might be the VZA effect considering that VZA increases at the northern parts in the area. Without angle conditions in the input parameters for the model, the difference becomes doubled at 312 nm presenting similar patterns with the difference in Fig. 12b. This indicates the angle effect can be emulated in the model by applying VZA and SZA as the input parameters, but it is not fully resolved especially for the radiances at shorter wavelengths.

[Figure]

*(a)*

*(b)*

***Figure 3*** *Same as Fig. 11 for the Defect 2 area.*

A closer inspection is performed to analyze the general spectral features over target wavelengths. For each defect area in Figs. 11-13, the collected spectra are divided into four groups depending on the scene brightness considering that ML radiances could have different systematic biases depending on the scenes. With the data, the mean difference is calculated for each wavelength. As found in Fig. 11, Fig. 14a shows that the ML radiances over dark scenes have the positive bias while brighter scenes have the negative bias. It is interesting that the scene dependence is only significantly found for Defect 1. Figure 14b indicates that the ML radiances are overestimated except for the very brighter scenes. It should be noted that the y-axis range of Fig.14b is wider than the figures for Defects 1 and 3. With the results, it can be deduced that the complicated atmospheric effects at the shorter wavelengths are difficult to be emulated and instrument artifacts such as stray light also would affect the reproducing process. Figure 14c shows relatively large difference at the spectral peaks, but generally the difference is smaller than 0.2%

[Figure]

*(a)*        *(b)*        *(c)*

***Figure 4*** *Mean difference between ML and GEMS radiances within the target area of (a) Defect 1, (b) Defect 2 and (c) Defect 3. Each color indicates the average for each quartile and Q1, Q2 and Q3 represent the first, second and third quartile, respectively. The difference is calculated as (ML-GEMS)/GEMS in percent.*

Besides the shorter wavelengths of Defect 2, mean ML radiance and the difference with GEMS radiances are presented by targeting Fraunhofer lines from 390 to 400 nm (see Fig. 14). The Ring effect caused by rotational Raman scattering can be found over the two peaks in Fig. 14a, which is generally known to be very small and largely affected by clouds (Joiner et al., 1995). Figure 14b shows that PCA-ANN reproduces the dominant features at the peaks very well on average within 0.6%, but it seems the difference increases with darker scenes where the Ring effect becomes stronger. This indicates that the ML radiances would need additional information to successfully reproduce the exact spectral features especially for the very small signals such as the Ring effect.

[Figure]

(a)                                         (b)

**Figure 5** *(a) Mean ML radiances (b) and the difference with GEMS raidances at the Fraunhofer lines for the Defect 2 area. Each color indicates the average for each quartile and Q1, Q2 and Q3 represent the first, second and third quartile, respectively. The difference is calculated as (ML-GEMS)/GEMS in percent.*

**3.2.2 PCA-based spectral analysis**

*As applied in the pre-processing step in our research, PCA is a very useful tool to capture the meaningful variances along the spectral direction and it has been widely used to retrieve environmental and surface properties (Horler and Ahern, 1986; Joiner et al., 2016; Li et al., 2013, 2015). To investigate further the spectral patterns, we apply PCA to GEMS radiances (except for bad pixels) at the target wavelengths (see Table 3) collected within each area in Fig. 11-13. With PCA, various spectral patterns are compressed and a spectrum can be projected to PC subspaces by multiplying with the constructed PC matrix (eigenvector matrix). This indicates that if a spectrum has disparate spectral patterns, the projected PCs would also have distinct values when comparing with the PCs of GEMS radiances. Figure 15 presents the results when projecting both GEMS and ML radiances with PCA. For the inspection, the Defect 3 area is presented which has the wider defective width along the north-south direction. Because the first PC scores represents mean radiances, the second PC are used for the analysis. As we assumed, bad pixels in Fig. 15a show disparate values because the spectral patterns of the interpolated spectra are inconsistent with GEMS radiances. The ML radiances in Fig. 15b show spatially homogenous PC scores which indicates that the machine learning methods could properly reproduce the dominant spectral patterns, in this case of the second PC.*

[Figure]

(a)                       (b)

**Figure 15.** *The second PC of (a) actual measurements and (b) reproduced spectra on the target area for Defect 3. The PC is scaled for clarity of presentation.*

*The dominant patterns for each PC are presented in Fig. 16 with GEMS radiances for the target wavelengths of Defects 1-3. Each color indicates the eigenvector of the first-sixth PCs which determines how each PC score of a spectrum contributes to the original spectrum. Li et al. (2015) verified that the leading PC scores from the UV/VIS backscattered radiation (shorter than 360 nm) are significantly correlated with dominant absorption features and surface properties. The trailing PC scores might be associated with instrument artifacts and other unresolved spectral features. Figure 16 shows that the first PC corresponds to the mean spectrum as discussed in Sect. 3.1.1 and the second-sixth PCs show dominant spectral patterns originated from absorption features of trace gases, surface properties and unresolved features. This indicates that the comparison of PC scores could provide the information on the similarity of the dominant patterns between ML and GEMS radiances as shown in Table 4 with the correlation coefficient. The results indicate that the mean spectral feature (the first PC) and some dominant patterns (the second and third PCs) could be well reproduced with the suggested models, but other spectral features such as the fourth PC for Defect 2 have difficulty obtaining valid information from input radiances for accurate reproduction. The magnitude of radiance from the major PCs except for the first PC might not be large considering that even the leading PCs have small explained variance ratio for hyperspectral data in UV/VIS spectrum. However, it would be enough to determine the exact spectral signals which are mostly related to the important information for the retrieval process.*

[Figure]

**Figure 6** *Eigenvector of the first-sixth PCs applied to GEMS radiances for the target wavelengths of (a) Defects 1, (b) Defect 2 and (c) Defect 3. All eigenvectors are scaled (min-max scaling) and shifted for clarity of presentation.*

**Table 2.** *Correlation coefficient of PC scores of reproduced and actual measurements for Defects 1-3.*

| *Defects* | *PC 01* | *PC 02* | *PC 03* | *PC 04* | *PC 05* | *PC 06* |
|-----------|---------|---------|---------|---------|---------|---------|
| *Defect 1* | *0.9999* | *0.9976* | *0.8172* | *0.9779* | *0.6846* | *0.6609* |
| *Defect 2* | *0.9999* | *0.8129* | *0.9876* | *0.4294* | *0.7035* | *0.5046* |
| *Defect 3* | *0.9999* | *0.9962* | *0.9787* | *0.6644* | *0.5399* | *0.2649* |

**Comment #2:**

**"Besides, how can the spectral sampling of input/output (0.1 nm) be finer than the original GEMS data (0.2 nm)? More detailed descriptions about this are recommended. Overall, I suggest this manuscript be reconsidered after major revisions."**

**Response #2:**
As for the spectral intervals of GEMS spectra for the training process, the response for the comment is addressed below as there is a similar comment in the following section.

**<Specific comments>**

- **Line 78: Please give the full names of the gaseous species (i.e., O3, SO2, NO2, and HCHO).**

  Corrected.

- **Line 82: The authors refer to each of ~700 east-west pixels as a "scan," but probably this term is not accurate. Isn't the whole ~700 pixels considered to be in one scan? Also, can GEMS cover the entire field of regard by one scan? It seems that is what the authors are implying.**

  The sentences have been revised as follows:

  "For earth measurements, GEMS measures the backscattered radiation from east to west about 700 times by moving a scan mirror and for each scan, totally 2048 pixels are obtained along the north-south direction. All measurements at each scan position are combined together to cover the full field of regard (FOR) of GEMS."

- **Line 84: Do the CCD pixel numbers presented here represent those for only photoactive pixels?**

  The provided pixel numbers are designed to be photoactive pixels. However, signals from some pixels at the edges of the CCD are known to be invalid, which are flagged as low quality pixels. The point has been added to the revised manuscript.

- **Line 89: The general description of the bad pixel detection method is informative. But how about presenting how long the GEMS integration time is (by adding another sentence)?**

  The integration time of GEMS is 69.996409 milliseconds. The information has been updated to the manuscript.

- **Line 99: This sentence sounds as if the results of 1-D interpolation were presented earlier, which is not true. How about rephrasing this sentence, using a verb like "imply" instead of "indicate"?**

  We agreed to the point. It has been corrected.

- **Line 104: The subject affected by the defective pixels is the quality of ozone retrieval, not the ozone properties themselves.**

  Indeed. It has been corrected.

- **Line 148: How can the spectral interval of input and output (0.1 nm) be narrower than that of original GEMS measurements (0.2 nm)? How are the GEMS measurement spectra sampled onto the finer grids? Please give more details here.**

The detailed description of the spectral interval of input and output has been added to Line 175:

*For the training process, each measured spectrum is linearly interpolated with the sampling interval of 0.1 nm, and radiances of each spectrum are divided into input and output radiances based on the specified spectral ranges in Table 2. The training datasets should be sampled at identical spectral grids and for that, each spectrum is interpolated in a pre-processing step. After the prediction, each replaced spectrum is reversely interpolated onto its original spectral grids. During the interpolation processes, intrinsic information a spectrum has could be lost, and thus finer spectral grids are applied to minimize interpolation errors by preserving radiances at more frequent interval than the original (about 0.2 nm).*

- **Line 149: Did you investigate how much the results changed when trained without solar zenith angle (SZA) and viewing zenith angle (VZA)? Please describe the impacts of including these variables.**

The impact of angle conditions as input has been analyzed and added to Line 175:

*Figure 5 presents the converging process of the PCA-ANN model for Defect 2 applying different optimizers with and without SZA and VZA conditions. The additions of the angle conditions as input parameters speed up the model convergence with smaller MSE because without the angle parameters, the information would be implicitly elicited during the optimization process. The model converges at 44, 98 and 33 epochs for Adam, SGD and RMSprop, respectively. Adam converges at the smallest MSE while the SGD converges with the highest MSE. RMSprop presents unstable loss for validation data and converges with higher MSE compared to Adam.*

[Figure]

*Figure 7 Training and validation losses for Defect 2 (a) with and (b) without the angle conditions as input parameters. The results are obtained with different optimizers such as Adam (black), SGD with the gradient clipping value of 0.5 (blue) and RMSprop (orange)*

- **Figure 5: The caption and the color bar title do not correspond. Which wavelength was used between 310 and 354 nm?**

Thanks for the correction. It is radiance at 310 nm and the caption has been corrected accordingly.

- **Line 264: How can we tell if spectra look "reasonable"? This statement is vague. Please consider changing Figs. 11-12 to include any reference (know, good, measured) spectra for the reconstructed parts.**

  The response for this comment is addressed in the previous section.

- **Line 269: I believe the term "noise" itself implies randomness, which would not necessarily be canceled in the normalized radiance. Please consider replacing the term with another, e.g., error, bias, artifact, etc.**

  Artifacts would be more proper expression, indeed. It has been updated.

- **Please consider re-writing the units in the figures as W cm–3 sr–1**

  Corrected.

- **Please consider minor English corrections below.**
  - **Lines 42, 49, 50, 100: affect to -> affect ?**
  - **Lines 109, 148, 184, 185, 199, 214, 221, 238, 241, 243, 250, 254, 258, 268: Defect -> Defects**
  - **Line 225: Fig. -> Figs.**
  - **Line 242: N-S -> North-South**
  - **Line 276: A period (.) missing between sentences**

  These comments have been addressed in the revised manuscript.

**Reference**

Horler, D. N. and Ahern, F. J.: Forestry information content of thematic mapper data, Int. J. Remote Sens., 7, 405–428, https://doi.org/10.1080/01431168608954695, 1986.

Joiner, J., Yoshida, Y., Guanter, L., and Middleton, E. M.: New methods for the retrieval of chlorophyll red fluorescence from hyperspectral satellite instruments: Simulations and application to GOME-2 and SCIAMACHY, Atmos. Meas. Tech., 9, 3939–3967, https://doi.org/10.5194/amt-9-3939-2016, 2016.

Li, C., Joiner, J., Krotkov, N. A., and Bhartia, P. K.: A fast and sensitive new satellite SO2 retrieval algorithm based on principal component analysis: Application to the ozone monitoring instrument, Geophys. Res. Lett., 40, 6314–6318, https://doi.org/10.1002/2013GL058134, 2013.

Li, C., Joiner, J., Krotkov, N. A., and Dunlap, L.: A new method for global retrievals of HCHO total columns from the Suomi National Polar-orbiting Partnership Ozone Mapping and Profiler Suite, Geophys. Res. Lett., 42, 2515–2522, https://doi.org/10.1002/2015GL063204, 2015.

---

## Author Response (AR1)

**Manuscript ID**: amt-2022-37

**Spectral replacement using machine learning methods for continuous mapping of Geostationary Environment Monitoring Spectrometer (GEMS)**

Yeeun Lee, Myoung-Hwan Ahn\*, Mina Kang, Mijin Eo

**General response to referees' comments**

We would like to thank Glen Jaross and the anonymous referee for the insightful comments and the detailed suggestions, which are significant to include some missed points not fully explained in the first draft. We have revised the manuscript considering the suggested comments and the referee's comments are presented in **bold**. In this document, we apply the figure and table numberings of the revised manuscript and the revised manuscript is presented in *italic*.

**Specific comments from Dr. Glen Jaross**

**Comment – Part A-1:**

**"I have difficulty understanding the goals of this paper. There seems to be a mismatch between what the authors are trying to achieve and the investigation they describe in the paper. The authors describe a problem on GEMS whereby "bad" pixels, i.e. missing pixel radiances, cause problems in the ensemble of measured radiances. … There is no mention of what criteria are used to assess the results."**

**Response A-1:**

We agree with the comments that the goals of this paper are not fully explained. Bad pixels of GEMS could cause distinct spatial discrepancy in GEMS Level 1 data and it definitely introduces errors in Level 2 products (or make impossible to derive the Level 2 products). Thus the primary goal of the current work is to evaluate the applicability of machine learning methods for the spectral gap filling, which could reduce bad pixel effects to both Level 1 and 2 products. For the purpose, bad pixel effects are discussed first (as suggested by the referee) and thus we have added the following part to Section 2.1, describing the effects of bad pixels to radiances and retrieved properties (i.e., ozone):

*The interpolation error could seriously affect the Level 2 products of which the spectral fitting windows are overlapped with bad pixel areas. For instance, cloud properties and aerosol effective height (AEH) of GEMS are retrieved from $O_2$-$O_2$ absorption bands around 477 nm (Choi et al., 2021; Kim et al., 2021) where the cluster of bad pixels is located (Defect 3). During the IOT, Defect 3 caused spatial discontinuity to the retrieved cloud and AEH distribution, which made the fitting window of the products modified to avoid bad pixel effects. The $O_3$ retrieval is also affected by Defect 2 (300-400 nm) as the spectral radiances within 300-380 nm provide major information for the $O_3$ retrieval of GEMS (Bak et al., 2019). The bad pixel effects in the Level 2 product are clearly shown in Fig. 2 which presents radiances at 312 nm and the retrieved total $O_3$ column of GEMS. Even though radiances at the certain wavelength are homogeneous with its surroundings (see Fig. 2b). The spectral patterns are not properly reproduced with the existed method (spatial interpolation) causing the distinct horizontal line in Fig.*

[Figure]

***Figure 1*** *Spatial distribution of (a) the GEMS total O₃ column and radiances at 312 nm with bad pixels (b) marked in dark gray and (c) reproduced with spatial interpolation. The measurements are on 10 March 2021 (06 UTC).*

**Comment – Part A-2:**

**"From an academic perspective the question of how well ML techniques can describe Earth backscattered radiances is an interesting one. If the authors approached this paper from that perspective they might provide a useful contribution to our ability to characterize the atmosphere through numerical techniques. But in doing so they must take a more rigorous approach to evaluating their radiance predictions.**

**The first thing the authors should do is to forget about the GEMS pixel defects. These are of no use in evaluating the efficacy of the technique since the true radiances remain unknown for these regions. Instead, choose regions of the detector where there are good measurements and treat them as missing for the purpose of deriving errors. There can be a variety of region shapes and sizes, including ones that look very similar to Defects 1, 2, and 3."**

**Response A-2:**

The points are also raised by another referee and we appreciate the suggestions. Following the suggestions, we targeted certain areas including each defect (Defects 1-3) and its surroundings (100-indices toward both north and south directions) where actual measurements (regarded as 'true') could be obtained. With the analysis, it is possible to quantitatively evaluate how the ML radiances differ with the true measurements. In summary, the direct comparison between measured and reproduced radiances shows that the dominant spectral patterns could be successfully reproduced even though the spectral patterns determined by very small signals such as the Ring effect would be insufficiently reproduced with the suggested methods. The spectral patterns are also evaluated by applying PCA to the measured and reproduced spectra, and the extracted spectral patterns are highly correlated for the first PC for Defect 1-3 regardless of the models which are consistent with the direct comparison results.

The results are added to the 'Results and discussion' (Section 3) which has been re-organized with 'Model selection' (Section 3.1) and 'Evaluation' (Section 3.2). The following part has been inserted to Section 3.2:

**3.2 Evaluation**

**3.2.1 Spatial and spectral inspection**

For the quantitative evaluation of the reproduced spectra, certain areas are targeted which include each defect (Defects 1-3) and its surroundings where actual measurements regarded as 'true' could be obtained. The evaluation is made with the data measured on 10 March 2021 (06 UTC), which are excluded for the model training. The center longitude of the areas is set to 128° E, which is identical with the sub-nadir longitude of GK-2B. Table 3 presents spectral ranges of Defects 1-3 and the target wavelengths for the analysis. Specifying the wavelengths for the analysis helps to specifically analyze the spectral patterns of absorption lines of trace gases and cloud properties.

*Table 1* The spectral range of Defects 1-3 and target wavelengths for the analysis. The third column presents GEMS retrieval products of which each fitting window is overlapped with Defects 1-3.

| Defect | Target wavelength | GEMS Level 2 product | Optimized model |
|---|---|---|---|
| 1 (400-500 nm) | 432-450 nm | CHOCHO, $NO_2$ | PCA-Linear |
| 2 (300-400 nm) | 312-360 nm | $O_3$, HCHO, $SO_2$, $NO_2$, aerosol optical depth | PCA-ANN |
| 3 (484-491 nm) | 484-491 nm | Cloud, AEH | PCA-Linear |

The measured and the reproduced radiances with machine learning methods are directly compared, which are hereafter referred to as GEMS radiances and ML radiances. In Figs. 11-13, each column shows GEMS, ML radiances and the difference while the first and second rows show the representative wavelengths showing the smallest and the largest difference, respectively. Figure 11 shows the comparison results of the Defect 3 area, which presents the best performance compared to the Defect 1 and 2 areas. The difference in Fig. 11 is within $\pm$ 0.5% because the spectral gap of Defect 3 is narrower than the counterparts of Defects 1-2. The narrower the spectral range of the output radiances is, the more abundant information could be obtained from the input radiances. 
[revised manuscript text omitted]

**Comment – Part A-3:**

**"But what is the real problem that the authors are trying to solve? In an instrument such as GEMS, designed to measure trace gas composition of the atmosphere through hyperspectral measurements, the goal is probably to produce trace gas products without spatial gaps caused by missing radiances. The authors do not discuss the issue of trace gas retrievals or other products derived from their predicted radiances. Is there any improvement at all in those products? …**

**The authors must also devise evaluation criteria that are more robust and quantitative than "these spectra look realistic." Since the goal for GEMS radiances is to derive atmospheric products such as trace gases, perhaps these trace gas retrievals can be used as the metric. Merely stating that predicted radiances agree on average with measured radiances to within X% ignores the subtle spectroscopic sensitivity of trace gases such as NO2, where the exact relationship between wavelengths is of utmost importance."**

**Response A-3:**

As presented in the previous section, we have evaluated the reproduced spectra by comparing with actual measurements and applied PCA to analyze spectral features of reproduced spectra. With the analysis, it was found that the machine learning models properly reproduce dominant spectral patterns for Defects 1-3 with only radiances from the rest part of spectra and angle conditions. However, the exact spectral features (<1%) determined by small signal (the important information) may be accurately reproduced only if the spectral range of output radiances are closer to the input radiances enough to obtain sufficient information from the input radiances (such as Defect 3). Also, it seems additional information would be needed to reproduce exact spectral features when the output radiances are overlapped with strong absorption or scattering lines. Considering the ultimate goal of measuring hyperspectral data, we agree with the referee's suggestion to apply the retrieval algorithms for evaluating the reproduced spectra. However, as the initial approach reproducing missing radiance of GEMS, we hope to evaluate the applicability of machine learning methods for the GEMS measurements, which contain meaningful information as well as instrument artifacts. As the referee pointed out, the

effect of reproducing spectra for the retrieval process is a necessary step and based on the findings in this research, we hope to investigate further the step in a follow-up study.

**Specific comments from the anonymous referee**

**Comment – Part B-1:**

*"I believe the demonstration of the method validity can be improved. In particular, the qualitative discussion about the performance of reproduction, described with Figs. 11-12, needs to be improved and more objective. Also, in addition to the prediction errors presented in Figs. 7–9, it is recommended to show how well the proposed methods reproduce known good spectra (i.e., actual measurements)."*

**Response B-1:**

Firstly, thanks for the valuable comments and suggestions. As the referee pointed out, we acknowledged that the result parts referred in the comment definitely need improvements and thus the Sections 2-3 have been greatly revised. The applied analysis in the first draft could not quantitatively prove the validity of the suggested methods. Following the two referee's recommendations, we targeted certain areas including each defective region (Defects 1-3) and its surroundings (100-indices toward both north and south direction) where actual measurements (regarded as 'true') could be obtained.

Considering that this comment is also related to the updated section for evaluating the reproduction results of the suggested methods (mostly Section 3), the detailed responses and revised part in the manuscript would be identical to **Response A-2**.

**Comment Part B-2:**

*"Besides, how can the spectral sampling of input/output (0.1 nm) be finer than the original GEMS data (0.2 nm)? More detailed descriptions about this are recommended. Overall, I suggest this manuscript be reconsidered after major revisions."*

**Response B-2:**

The detailed description of the spectral interval of input and output has been added to Line 159 (w/o track changes):

*The datasets for the models should be sampled at identical spectral grids and for that, each spectrum is interpolated in a pre-processing step and after the reproduction, the spectra are reversely interpolated onto its original spectral grids. Considering that the intrinsic information a spectrum has could be lost during the interpolation processes, the finer spectral grids (0.1 nm) are adopted for the model to minimize interpolation errors by preserving radiances at more frequent intervals than the original grids.*

**Comment Part B-3:**

- **Line 78: Please give the full names of the gaseous species (i.e., O3, SO2, NO2, and HCHO).**

  Corrected.

- **Line 82: The authors refer to each of ~700 east-west pixels as a "scan," but probably this term is not accurate. Isn't the whole ~700 pixels considered to be in one scan? Also, can GEMS cover the entire field of regard by one scan? It seems that is what the authors are implying.**

The sentences have been revised as follows:

"For earth measurements, GEMS measures the backscattered radiation from east to west about 700 times by moving a scan mirror and for each scan, totally 2048 pixels are obtained along the north-south direction. All measurements at each scan position are combined together to cover the full field of regard (FOR) of GEMS."

- **Line 84: Do the CCD pixel numbers presented here represent those for only photoactive pixels?**

The provided pixel numbers are designed to be photoactive pixels. However, signals from some pixels at the edges of the CCD are known to be invalid, which are flagged as low quality pixels. The point has been added to the revised manuscript.

- **Line 89: The general description of the bad pixel detection method is informative. But how about presenting how long the GEMS integration time is (by adding another sentence)?**

The integration time of GEMS is 69.996409 milliseconds. The information has been updated to the manuscript.

- **Line 99: This sentence sounds as if the results of 1-D interpolation were presented earlier, which is not true. How about rephrasing this sentence, using a verb like "imply" instead of "indicate"?**

We agreed to the point. It has been corrected.

- **Line 104: The subject affected by the defective pixels is the quality of ozone retrieval, not the ozone properties themselves.**

Indeed. It has been corrected.

- **Line 148: How can the spectral interval of input and output (0.1 nm) be narrower than that of original GEMS measurements (0.2 nm)? How are the GEMS measurement spectra sampled onto the finer grids? Please give more details here.**

As for the spectral intervals of GEMS spectra for the training process, the response for the comment has been addressed in the previous section.

- **Line 149: Did you investigate how much the results changed when trained without solar zenith angle (SZA) and viewing zenith angle (VZA)? Please describe the impacts of including these variables.**

The impact of angle conditions as input has been analyzed and added to Line 177 (w/o track changes):

*Figure 5 presents the converging process of the PCA-ANN model for Defect 2 applying different optimizers with and without SZA and VZA conditions. The additions of the angle conditions as input parameters speed up the model convergence with smaller MSE because without the angle parameters, the information would be implicitly elicited during the optimization process. The model converges with angle conditions at 44, 98 and 33 epochs for Adam, SGD and RMSprop, respectively. Adam converges at the smallest MSE while the SGD converges with the highest MSE. RMSprop presents unstable loss for validation data and converges with higher MSE compared to Adam.*

[Figure]

*Figure 8* Training and validation losses for Defect 2 (a) with and (b) without the angle conditions as input parameters. The results are obtained with different optimizers such as Adam (black), SGD with the gradient clipping value of 0.5 (blue) and RMSprop (orange).

- **Figure 5: The caption and the color bar title do not correspond. Which wavelength was used between 310 and 354 nm?**

  Thanks for the correction. It is radiance at 310 nm and the caption has been corrected accordingly.

- **Line 264: How can we tell if spectra look "reasonable"? This statement is vague. Please consider changing Figs. 11-12 to include any reference (know, good, measured) spectra for the reconstructed parts.**

  The response for this comment is addressed in the previous section.

- **Line 269: I believe the term "noise" itself implies randomness, which would not necessarily be canceled in the normalized radiance. Please consider replacing the term with another, e.g., error, bias, artifact, etc.**

  Artifacts would be more proper expression, indeed. It has been updated.

- **Please consider re-writing the units in the figures as W cm–3 sr–1**

  Corrected.

- **Please consider minor English corrections below.**
  - **Lines 42, 49, 50, 100: affect to -> affect ?**
  - **Lines 109, 148, 184, 185, 199, 214, 221, 238, 241, 243, 250, 254, 258, 268: Defect -> Defects**
  - **Line 225: Fig. -> Figs.**
  - **Line 242: N-S -> North-South**
  - **Line 276: A period (.) missing between sentences**

  These comments have been addressed in the revised manuscript.

---

## Author Response (AR2)

**Manuscript ID**: amt-2022-37

**Spectral replacement using machine learning methods for continuous mapping of Geostationary Environment Monitoring Spectrometer (GEMS)**

Yeeun Lee, Myoung-Hwan Ahn*, Mina Kang, Mijin Eo

**General response to referees' comments – major revision**

It is highly appreciated for the detailed comments of Glen Jaross to greatly improve the revised manuscript. We tried to reflect the referee's comments as much as possible and the manuscript has been reorganized to take better into account the referee's comments. More specifically, the revised version includes the impact of ANN-filled radiance to the Level 2 products such as cloud and ozone. One thing to be noted is that the data used in this paper have been updated with the operational data (Level 1C) for the analysis of retrieval results. In this reply, the referee's comments are repeated in blue, our responses to the specific comment are given in red, and the revised manuscript is presented in *italic*.

**Specific comments from Dr. Glen Jaross**

**1. The authors have partially addressed my concerns expressed after my initial review. In their Version 2 they offer a quantitative evaluation of ANN radiance errors, which addresses one of my concerns.**

⇨ Thanks a lot.

I repeat my earlier criticism that **the authors have not clearly stated the objective of this paper**. In the introduction the authors identify the fundamental problem they are trying to address: Level 2 products have spatial gaps due to bad pixels. Logically, the next step is to ask the question: what is the best approach to filling these gaps? The authors do not ask this question but instead proceed to discuss a radiance replacement approach in the Level 1 product. In my opinion there are multiple solutions that will better fill Level 2 gaps than radiance replacement in the Level 1 product. The authors should acknowledge there are alternative solutions to this problem and tell the readers this paper describes the investigation of one technique.

⇨ Okay. We acknowledged the objective of the study and its justification was not fully explained in the previous version of the manuscript. The starting point of this current work was to mitigate aesthetic annoyance that the released image would produce but we admitted that ultimately, filling the gaps in Level 2 products would be the final goal for the issue. It is true that the annoyance could be mitigated by other approaches such as simple interpolation of the Level 2 products (would be very efficient for the products having a smooth spatial variation). However, we wanted to touch upon a bit more the fundamental issue, "could the machine learning approach provide a new insight on the missing radiances caused by the bad pixels?" We didn't expect any algorithm would provide full content of information that only the actual measurements data could provide. However, some cases, the available nearby data (either in space or in spectral domain) could fill the gap introduced by the bad pixels with sufficient accuracy that could be used for further application. Furthermore, the carefully prepared machine learning algorithm could provide some information even on the rather complicated spectral range (as described below).

⇨ Reflecting our response, the revised part of manuscript is (or are);

***Section 1 Introduction (Lines 50-64)***

*Especially, when a scene on the Earth dramatically changes, discontinuity caused by the interpolation becomes larger. This effect causes spatial discontinuity in Level 1B data and retrieved properties (Level 2) by affecting retrieval processes with contaminated spectral features.*

*As a way of filling in the spatial gaps, this study approaches the underlying problem by focusing on radiances with spectral replacement using machine learning methods. The spatial gaps found in Level 2 data can be filled in with various methods (e.g. variogram, empirical orthogonal functions or mathematical filters) and for each Level 2 product, there will be a more suitable method using multiple sources of information and distribution characteristics (Fang et al., 2008; Guo et al., 2015; Katzfuss and Cressie, 2011; Llamas et al., 2020; Yang et al., 2021). In this regard, this research places more emphasis on efficiency and further application of the approach because improving erroneous spectral features can be an efficient way to solve the issue for all products and also has the potential to be applied to various measurement issues of hyperspectral data. For that, further questions to be investigated here are whether non-linear relations could be accurately emulated with machine learning methods and input radiances have valid information for retrieval processes. For the investigation, cloud and ozone retrievals are performed with the reproduced spectra of GEMS to evaluate the effectiveness of the suggested approach and its limitations.*

\

**2. The authors attempt a quantitative evaluation of the replacement radiances for the GEMS pixel gaps, but the reader is left wondering how important the residual errors are. A correlation coefficient of 0.82 in PC2 or PC3 doesn't sound good, but what does that really mean in terms of product performance? The paper contains some discussion regarding the Ring Effect, but otherwise little is said about how useful ANN-based radiances are. The very big question remaining is, is there any improvement in Level 2 performance and does that performance warrant further pursuing the ANN technique? The authors may argue that this is the subject of a future paper. If so, this should be made clear in both the stated objectives and in the conclusions.**

⇨ The straightforward approach to answer the usefulness of ANN-based radiances is analyzing the impact of reproduced spectra on the Level 2 products. As we suggested in the previous reply and the referee pointed out, the task will incur a new research work which will certainly be carried out in the near future. However, to check the possibility and ensure future work, we analyzed the effects of spectral replacement for the Level 2 retrieval processes (in Sect. 3.3.1) as suggested, especially for ozone and cloud properties with the help of the GEMS Level 2 algorithm team. Considering that the PCA results still provide the information on how qualitatively successful the spectral replacement is for different spectral regions in terms of spectral relations, the PCA part is reorganized in Sect. 3.2.2 to support the retrieval results presented in the following section.

⇨ Reflecting our response, the revised parts of the manuscript are;

*Section 3.2.2 PCA-based analysis (Lines: 294-305)*
*As presented in Table 4, comparing PC scores provides qualitative information on the effectiveness of the suggested method. The results show that the mean spectral pattern (the first PC) and some dominant patterns can be sufficiently reproduced with the suggested models, but other spectral features such as the third PC for Defect 1 or the second PC for Defect 2 have difficulty obtaining valid information from input radiances for accurate reproduction. The interesting finding is that only Defect 3 shows high correlation coefficients over 0.95 for the leading PCs having higher explained variance ratios. Each PC except the first one may contribute to a small extent to total radiances given the explained variance ratio. However, it also could be enough to determine subtle spectral patterns important for retrieval processes. The effectiveness of spectral replacement for each spectral region could be glimpsed in the results, which will be discussed further in the following section with retrieval processes.*

*Table 1 Correlation coefficients of PC scores of GEMS and ML radiances and explained variance ratios of GEMS radiances for each target region in Fig. 8-10.*

| Defects | Factor | PC 01 | PC 02 | PC 03 | PC 04 | PC 05 | PC 06 |
|---|---|---|---|---|---|---|---|
| 1 | Correlation coefficient | 0.9999 | 0.9976 | 0.8172 | 0.9779 | 0.6846 | 0.6609 |
| 2 | | 0.9999 | 0.8129 | 0.9876 | 0.4294 | 0.7035 | 0.5046 |
| 3 | | 0.9999 | 0.9962 | 0.9787 | 0.6644 | 0.5399 | 0.2649 |
| 1 | Explained variance ratio [%] | 99.9905 | 0.0071 | 0.0007 | 0.0006 | 0.0001 | 0.0001 |
| 2 | | 99.9524 | 0.0268 | 0.0141 | 0.0019 | 0.0012 | 0.0005 |
| 3 | | 99.9954 | 0.0038 | 0.0003 | 0.0001 | 0.0001 | 0.0001 |

*Section 3.3.1 Cloud and ozone retrieval (Lines 307-330)*
*In the previous section, it was found that the overall prediction error is about 5% except for the ozone absorption lines and dominant spectral patterns can be successfully reproduced with the suggested method. The next question to be discussed is whether the reproduced spectral features are applicable to the retrieval process. Even if the trained models accurately reproduce an absolute value at each wavelength, the Level 2 retrieval could be unsuccessful if non-linear relations are*

*too elusive to be properly emulated with the model. The radiances at O2-O2 absorption lines related to Defect 3 has the smallest prediction error of 0.5% and we checked that cloud information with the fitting window in 460.2-490.0 nm can be successfully retrieved with the reproduced spectra in Fig. 8. The difference of cloud centroid pressure retrieved with ML and GEMS spectra is about 1% on average for normal measurements but the cloud properties retrieved with ML spectra have weak stripped features. The spectral range of Defect 3 is very narrow and thus the input radiances provide enough information for successful spectral replacement and the retrieval process.*

*The replaced radiances at ozone absorption lines showed high prediction error in the previous section. For the qualitative investigation of the effect, the reproduced spectra presented in Fig. 10 are applied to the ozone retrieval of GEMS. Figure 13 shows total ozone column density with unlagged bad pixel area for the comparison of spatial discontinuity. As previously mentioned in Sect. 2.1.2, the ozone properties retrieved with measured GEMS spectra show distinct spatial discontinuity over the bad pixel area as shown in Fig. 13a and the discontinuity is somewhat reduced in Fig. 13b with ML spectra. However, the retrieved properties show different spatial distribution patterns when comparing the surrounding areas which have true measurements. It seems the ozone properties are underestimated especially for higher radiances in Fig. 13b and the stripping features found in Fig. 10 may affect the retrieval process considering the features are also found in Fig. 13b. It is also clear that the angle conditions provide important information for the retrieval because without the conditions, the retrieval results show unrealistic features with higher variance. The results indicate that the spatial distribution can be approximately matched for the ozone properties retrieved with measured (true) and reproduced spectra, but it seems limitation still remains to get an exact retrieval value.*

[Figure]

*Figure 1* *Spatial distribution of total ozone column density retrieved with (a) GEMS and (b) ML radiances presented in Fig.11. The GEMS spectra are measured on 10 March 2021 (06 UTC).*

**3. My general impression of the paper is it spends too much time considering the specific GEMS data cube gaps, especially in the paper section concerning evaluation. As the authors point out in their conclusions, the radiance replacement approach may be useful for other instruments, and those instruments will almost certainly not share the same gaps as GEMS. The readers would be better served if the authors investigate a variety of data gaps and demonstrate the efficacy of the ANN replacement method in each. The GEMS gaps can be a subset of those tested. Such an approach ties in better with a modified objective of the paper, where the emphasis is the strengths and weaknesses of ANN radiances under a variety of circumstances.**

⇨ It is quite an interesting suggestion. By doing so, we may prepare for the bad pixel issues GEMS would face in the coming years. On the other hand, the data gaps identified in the current GEMS data could give representative examples of bad pixels that a hyperspectral instrument would have. One is the complex ozone absorption bands, another one is the rather smooth spectral bands in 400-500 nm, while the other is the rather flat and highly correlated (with neighboring wavelengths) bands. Thus, the characteristics revealed with the three gaps could represent general data gaps we would expect from a hyperspectral instrument in the UV/VIS spectral range. However, we fully agreed that the input conditions were not fully investigated because of the retrained conditions, and thus we have inserted the spectral analysis following the retrieval results for the cause analysis of retrieval results and further application of the method.

⇨ The revised parts of the manuscript are;

**3.3.2 Cause analysis for further application (Lines 331-378)**

[revised manuscript text omitted]

**4. If the objective of the paper is to propose a solution for the GEMS missing pixels, I believe the discussion presented in Version 2 of this paper is incomplete. But if the objective is to describe a radiance replacement technique that is the basis for further investigation, the discussion presented in this current version is appropriate. In that case the authors should spend less time analyzing the specific GEMS data gaps, though they can be discussed as the impetus for the investigations.**

⇨ Again thanks for the raising the point. The ultimate goal is to increase usefulness of GEMS data for a longer time period, at least for designed lifetime of 10 years under the new operational environment of geostationary. The current study shows that the gap filling (in level 1) over the longer wavelengths are quite feasible and reliable, while it still haves limitations for strong absorption bands which may provide the reasons why we need actual observation data over such spectral bands. However, accumulation of observation and auxiliary data along with an improved nonlinear algorithm, the shortage of gap filling over the absorption bands could be improved, we hope.

⇨ As introduced in the earlier part of our response, the introduction part as well as overall sections of the manuscript have been revised by relating the resulting problem of bad pixels (i.e. spatial gaps in Level 2) and the method we chose to deal with (i.e. spectral replacement) with the broadened perspective based on the suggestion by applying ozone and cloud retrievals with the reproduced spectra and analyzing spectral gaps with different input conditions.

⇨ Reflecting our response, the revised parts in the manuscript are;

*Section 4: Conclusions (Lines: 380-415)*

[revised manuscript text omitted]

---

## Author Response (AR3)

**Manuscript ID**: amt-2022-37

**Spectral replacement using machine learning methods for continuous mapping of Geostationary Environment Monitoring Spectrometer (GEMS)**

Yeeun Lee, Myoung-Hwan Ahn*, Mina Kang, Mijin Eo

**General response to the reviewer' comments – minor revision**

Again, we appreciate the efforts made by the reviewer to clarify the objective of the manuscript and improve the quality of the manuscript. We have tried to reflect the comments and suggestions as much as possible. Accordingly, the introduction and conclusion parts are revised along with the editorial works to concisely deliver the information of the research. In this document, the reviewer's comments are repeated in blue, our responses to the specific comment are given in red, and the revised manuscript is presented in *italic*.

<Major comments>

**1. Research objective**

"The authors have cited several papers describing earlier attempts at replacing bad pixels. The existence of other attempts does not mean such replacement is right for GEMS data products. One might argue that in a situation where a geophysical parameter is over-determined by the available data it is possible to accurately predict and replace missing data. But by the same argument those missing data were not really needed to determine the geophysical parameter. **Other than stating that bad pixels should be replaced, the authors offer no explanation of how replaced pixels can improve a GEMS product.** After all, these are not measurements. If the purpose of a product is to report measurements, why should it report something other than measurements? Gap filling might allow a product to be generated in locations where one was previously unavailable, **but what is the value of that product to the broader user community?** In all likelihood they will treat the synthetic data equally alongside the real data. In that case pixel replacement will have done a disservice to the science. …"

"One example I can think of to support pixel replacement is that of reflecting surface pressure (a.k.a. cloud height). The authors have indicated that Defect 3 affects the primary $O_2$-$O_2$ absorption used to derive an altitude. ML or PCA may be able to identify a correlation between $O_2$-$O_2$ line depth and rotational Raman broadening at other, unaffected wavelengths, and thereby transfer the cloud height information back into a synthetic $O_2$-$O_2$ line. This would relieve the GEMS program from having to develop an entirely new cloud height replacement algorithm based on the RRS signal. I find this a tenuous argument at best, but the authors may choose to cite examples along these lines."

⇨ Thanks again for giving us a chance to revisit the manuscript. We expect that the points raised by the reviewer are clearly presented in the revised manuscript. In section 1, the advantages of replacing Level 1B data have been inserted to state plainly how the replaced pixels improve the GEMS products and what the advantages of the approach are. Because GEMS is a geostationary satellite sensor, bad pixel effects cause a permanent measurement gap for certain areas in the GEMS field of regard. As the reviewer pointed out, the reproduced values could not provide the information possibly obtained from actual measurements. On the other hand, one may need the most probable values likely measured by GEMS for various reasons (practical or scientific) for the information gaps. Here we tried to evaluate the applicability of machine learning in this regard presenting the analysis results and limitations for the issue. The suggested example ($O_2$-$O_2$ & rotational Raman scattering lines for cloud height retrieval) also has been included as one of advantages as it represents the effectiveness of spectral replacement.

**#2. Introduction (Section 1)**

"… However, **the justification is scattered throughout the introduction section. It reads like it was slipped in as an afterthought.** What this paper still lacks is a clear, up front statement of "this is why we are investigating ML techniques specifically." **Arguments about ease of implementation and benefits to multiple atmospheric parameters should appear at or near the beginning of the Introduction section**. **The authors should state clearly that the purpose of their investigation was to explore how well ML works to describe missing pixel content and not to find the best or most accurate pixel replacement method** (for example, Level 2 product assimilation followed by radiative transfer modeling and instrument modeling might prove more effective)."

⇨ Thanks. The reason why we try to provide the most probable Level 1B radiances (rather than the Level 2 properties) with machine learning has been revised considering the reviewer's concerns.

⇨ Reflecting our responses for Sect. 1, the revised part of manuscript is:

*Section 1 Introduction (Lines 36-56)*

*… The impact of bad pixels to the GEMS data products is obvious because the given areas affected by bad pixels cannot provide any measured information. It causes spatial discontinuity in Level 1B data and retrieved properties (Level 2) by affecting retrieval processes with contaminated spectral features.* ***The defective region is not large so far, but the area could be enlarged as time goes by*** *(Kieffer, 1996)* ***and the missing areas may increase possibly including scientifically important regions especially for environmental monitoring.***

***Because there is a constant measurement gap for certain areas in the GEMS field of regard (FOR), one could need alternative information for the areas for practical or scientific reasons.*** *To supplement the information and investigate the applicability of machine learning, this study focuses on replacing the Level 1B radiances using spectral relations with simple machine learning methods.* ***One of advantages of replacing Level 1B data (not the Level 2) is that improving spectral features can be an efficient way to solve the bad pixel issue for all Level 2 products.*** *The proposed approach places more emphasis on efficiency and further applicability of machine learning, even though the spatial gaps in Level 2 data can be filled with a suitable method for each product with higher accuracy (e.g., variogram or mathematical filters)* (Fang et al., 2008; Katzfuss and Cressie, 2011; Guo et al., 2015; Llamas et al., 2020; Yang et al., 2021). ***Another advantage is that the approach helps the current retrieval algorithms avoid bad pixel effects without further development.*** *The GEMS cloud height retrieval algorithm, for instance, had to modify the fitting window during the IOT because the targeted O2-O2 absorption lines (around 477 nm) are affected by bad pixels. The proposed approach, however, has the potential to reproduce the O2-O2 absorption features with the information from unaffected wavelengths (e.g., rotational Raman scattering lines) by applying spectral replacement. If it is successful, the retrieval can avoid bad pixel effects without further algorithm development. The main question to be answered for that is whether non-linear spectral relations could be effectively emulated with spectral replacement using machine learning techniques.*

**Section 3.3.1**

"The authors state that cloud height retrievals from Defect 3 appear to have an accuracy 1% when comparing measured and ML values. They also state that this success is a consequence of the spectrally narrow defect. ML is more likely to predict the correct spectra over a narrow range of wavelengths. Is this really the correct logical conclusion? Is it possible the good agreement is caused by natural spatial homogeneity in cloud heights. Cloud heights do not vary much within small spatial regions, so one might expect such agreement regardless of what pixel replacement technique is used. It's hard to believe that ML possesses enough information to accurately predict a high cloud that is surrounded by uniformly low clouds. Showing that it is capable of doing so will demonstrate that the ML technique has some real value."

⇨ Because the spectral replacement we applied only uses spectral relations of radiances in a spectrum, the spatial homogeneity of the retrieval properties hardly affects the replacement results. However, we understand the final statement in the section for cloud retrieval might mislead the point as the reviewer pointed out. The section has been revised and Fig. 12 has been inserted in the revised version for the demonstration as commented.

⇨ Reflecting our responses, the revised part of manuscript is:

*Section 3.3.1 (Lines 283-296)*

*In the previous section for radiances, the overall prediction error with the suggested method is about 5% except for ozone absorption lines. The next question is whether the reproduced spectral features are applicable to retrieval processes. Even if the trained models accurately reproduce radiances at each wavelength, the Level 2 retrieval could be unsuccessful if non-linear relations are too elusive to be properly emulated with the model.* **To prove this, we performed the cloud retrieval with the fitting window in 460.2-490.0 nm containing bad pixels.** *The replaced radiances at O2-O2 absorption lines related to Defect 3 have the smallest error of 0.5% and the retrieval is successful as shown in Fig. 12. Without the replacement, the retrieved cloud centroid pressure showed unrealistic values on bad pixel areas. Figure 12 presents cloud centroid pressure retrieved with ML and GEMS spectra by zooming in defect-free areas to analyze cloud distribution. The difference of cloud centroid pressure between Figs.12a and 12b is about 1% on average while the cloud properties of ML spectra have weak stripping features. The spectral range of Defect 3 is very narrow within the fitting window and thus the replacement errors could be small enough not to cause additional retrieval errors.*

[Figure]

[Figure]

***Figure 1*** *Spatial distribution of cloud centroid pressure retrieved with (a) GEMS and (b) ML radiances presented in Fig.7. The GEMS spectra are measured on 10 March 2021 (06 UTC).*

**Figure 14**

"The various colored lines in this figure need more explanation, either in the text or in the figure caption."

⇨ The section in Sect. 3.3.2 and the figure caption has been revised accordingly.

*Section 3.3.2 (Lines 319-327)*

*Figure 14 presents mean absolute errors of reproduced radiances for ozone absorption and Fraunhofer lines with four different input conditions: 1-2) including each near side (within 20 nm) from the output spectral regions (A and B for the left and the right side, respectively); 3) including both near sides of wavelengths (A and B); and 4) all wavelengths in 300-500 nm except for A, B and the output spectral region. Each input case is plotted in Fig 14 with the color of red, sky blue, blue and black line, respectively.* Results show that prediction errors increase at the spectral peaks and overall error patterns differ for different input conditions. As assumed, the errors are higher with farther input spectral bands from the output spectral region. Figure 14a clearly shows that the insufficient information from the input data may cause large errors for radiances at shorter wavelengths as well as the ozone retrieval. Figure 14b also presents that each input case has a different level of information which could determine the accuracy of spectral replacement especially for the weak scattering features.

**Section 4**

"This section reads more like a Summary of what has already been discussed in previous section rather than actual Conclusions. Please spend more time describing what works well and what does not work well, and suggest explanations for this performance."

The discussion starting at line ~400 is good, and I would like to the authors to expand this some more. The authors conclude that ML is capable of filling spatial gaps and narrow spectral gaps, but not larger spectral gaps. A little more insight into why this is the case will be appreciated. **It's important for the reader to understand the defect situation where further development of the ML technique might yield better results, and the situations where no amount of additional development is likely to improve the results.** The authors may wish to offer suggestions for alternative gap-filling techniques in this latter situation."

⇨ The following part has been inserted in the revised version. We hope the part could effectively deliver important findings we could provide for readers.

*Section 4 (Lines 378-390)*

*Further investigation reproducing Fraunhofer lines and ozone absorption lines helps conclude the benefits and limitations of the approach as follows: 1) The closer the input and output wavelengths are, the smaller its reproduction error becomes. This is because radiances at adjacent wavelengths have a high possibility containing common information valid for the replacement. Even though the condition is not satisfied, approximate spatial patterns could be obtained but the accuracy is not guaranteed for both radiances and retrieval properties. 2) The input radiances should be carefully selected because machine models (especially ANN) are vulnerable to outliers or erroneous input radiances. If one adopts more complex models, the importance of the selection would increase. 3) Errors coming from instrument artifacts such as the stripping feature could be propagated with the method as it seems the feature is not properly emulated in the model so far. 4) Finally, low radiances could have higher uncertainty even when using the spectral*

*information as much as possible. GEMS is the environmental sensor and thus may provide useful information with clear sky conditions. Considering this, additional information would be needed if one pursues very high retrieval accuracy with the replaced spectra. In this regard, combining the external information together with the spectral components would be the next step for developing the approach. Additionally, the research adopts very simple machine learning models which also can be updated further.*

**General**

"I find the text rather 'wordy', and have difficulty in places understanding the point the authors are trying to convey. It will help if the paper contains simple and clear messages. "We did the following because of X, Y, and Z." Or, "we found the method worked best when we included the data between x1 and x2." I know the topic is complex and there is no simple way to describe some of the work that was done, but the job of the author is to condense a complicated subject into something the readers can follow and digest."

⇨   Indeed. We have tried to revise the overall contents as concise as possible especially for Sects. 1 and 3. Please note that some parts have been deleted or reorganized considering that the parts have repetitive information and need refined explanation.

**Reference**

Fang, H., Liang, S., Townshend, J. R., and Dickinson, R. E.: Spatially and temporally continuous LAI data sets based on an integrated filtering method: Examples from North America, Remote Sens Environ, 112, 75–93, https://doi.org/10.1016/J.RSE.2006.07.026, 2008.

Guo, L., Lei, L., Zeng, Z. C., Zou, P., Liu, D., and Zhang, B.: Evaluation of spatio-temporal variogram models for mapping Xco2 using satellite observations: A case study in China, IEEE J Sel Top Appl Earth Obs Remote Sens, 8, 376–385, https://doi.org/10.1109/JSTARS.2014.2363019, 2015.

Katzfuss, M. and Cressie, N.: Spatio-temporal smoothing and EM estimation for massive remote-sensing data sets, J Time Ser Anal, 32, 430–446, https://doi.org/10.1111/J.1467-9892.2011.00732.X, 2011.

Kieffer, H. H.: Detection and correction of bad pixels in hyperspectral sensors, in: Hyperspectral Remote Sensing and Applications, 93–108, https://doi.org/10.1117/12.257162, 1996.

Llamas, R. M., Guevara, M., Rorabaugh, D., Taufer, M., and Vargas, R.: Spatial Gap-Filling of ESA CCI Satellite-Derived Soil Moisture Based on Geostatistical Techniques and Multiple Regression, Remote Sensing 2020, Vol. 12, Page 665, 12, 665, https://doi.org/10.3390/RS12040665, 2020.

Yang, M., Khan, F. A., Tian, H., and Liu, Q.: Analysis of the Monthly and Spring-Neap Tidal Variability of Satellite Chlorophyll-a and Total Suspended Matter in a Turbid Coastal Ocean Using the DINEOF Method, Remote Sensing 2021, Vol. 13, Page 632, 13, 632, https://doi.org/10.3390/RS13040632, 2021.